# Real-Time Human-In-The-Loop Simulation with Mobile Agents, Chat Bots, and Crowd Sensing for Smart Cities

**DOI:** 10.3390/s19204356

**Published:** 2019-10-09

**Authors:** Stefan Bosse, Uwe Engel

**Affiliations:** 1Faculty Computer Science, University of Koblenz-Landau, 56070 Koblenz, Germany; 2Department of Social Science, University of Bremen, 28359 Bremen, Germany; uengel@uni-bremen.de

**Keywords:** simulation, agent-based modelling, mobile agents, crowd sensing, smart traffic control, social interaction

## Abstract

Modelling and simulation of social interaction and networks are of high interest in multiple disciplines and fields of application ranging from fundamental social sciences to smart city management. Future smart city infrastructures and management are characterised by adaptive and self-organising control using real-world sensor data. In this work, humans are considered as sensors. Virtual worlds, e.g., simulations and games, are commonly closed and rely on artificial social behaviour and synthetic sensor information generated by the simulator program or using data collected off-line by surveys. In contrast, real worlds have a higher diversity. Agent-based modelling relies on parameterised models. The selection of suitable parameter sets is crucial to match real-world behaviour. In this work, a framework combining agent-based simulation with crowd sensing and social data mining using mobile agents is introduced. The crowd sensing via chat bots creates augmented virtuality and reality by augmenting the simulated worlds with real-world interaction and vice versa. The simulated world interacts with real-world environments, humans, machines, and other virtual worlds in real-time. Among the mining of physical sensors (e.g., temperature, motion, position, and light) of mobile devices like smartphones, mobile agents can perform crowd sensing by participating in question–answer dialogues via a chat blog (provided by smartphone Apps or integrated into WEB pages and social media). Additionally, mobile agents can act as virtual sensors (offering data exchanged with other agents) and create a bridge between virtual and real worlds. The ubiquitous usage of digital social media has relevant impact on social interaction, mobility, and opinion-making, which has to be considered. Three different use-cases demonstrate the suitability of augmented agent-based simulation for social network analysis using parameterised behavioural models and mobile agent-based crowd sensing. This paper gives a rigorous overview and introduction of the challenges and methodologies used to study and control large-scale and complex socio-technical systems using agent-based methods.

## 1. Introduction

The key concept of this work is the consideration of humans as sensors and the fusion of real and virtual worlds, in particular, virtual worlds in terms of social simulation, providing a closed-loop simulation. The outcome of simulations performed in real-time can be feedback to real worlds, e.g., to control crowd behaviour and flows, thus considering humans as actors. This human-in-the-loop simulation methodology enables a better understanding of crowd behaviour in real worlds and the opportunity to influence real worlds by simulation. This concept is highly interdisciplinary and is a merit of social and computer science if human sensor data will be coupled with social interaction and networking models. Social interaction has a high impact on the control of complex environmental and technical systems, which should be addressed in this work.


*The novelty of this work is the seamless fusion of real and virtual worlds by using mobile agents and a unified agent platform that can be deployed in strong heterogeneous digital network environments and in simulation. The real world is sensed by mobile crowd sensing techniques (MCWS) providing input for an agent-based simulator representing the virtual world. Mobile agents are used to bridge both worlds and to provide the loose coupling required in heterogeneous and mobile environments with ad-hoc connectivity and operation on a wide range of host platforms with varying software versions.*


The approach of an augmented agent-based social simulation is a powerful way to study large-scale social and socio-technical interactions. The strength of the framework lies in the potential inclusion of environmental and real-life interactions and providing feedback to real-world environments and humans. This agent platform, when used as the core component of both the MCWS and the simulator enables ubiquitous participation and interaction.

Different use-cases show the strength of the extended simulation framework using self-organising parameterised behaviour and interaction models with parameter sets derived by crowd sensing. The crowd sensing is performed by mobile agents that are used to create individual parameterised digital twins in the simulation from surveys performed by real humans (with respect to the social interaction model and mobility).

The following introduction gives an overview of agent-based modelling, simulation, and crowd sensing in the context of social and computer science, and enlightens the key concepts of agent-based human-in-the-loop simulation. After the Introduction, the role of intelligent systems and the relation to system control are discussed in Section 2, and modelling of socio-technical systems is discussed in Section 3. In Section 4 and Section 5, the proposed simulation architecture and the workflow are described. Section 6 describes the agent-based real-world sensing using crowd sensing techniques. In Section 7, a parameterised social interaction model is discussed, and Section 8 introduces a crowd mobility and traffic model. Finally, in Section 9 three use-cases demonstrate the suitability and utilisation of the augmented simulation approach in different fields of application.

### 1.1. Preliminary Notes on the Involved Computational Social Science (CSS) Perspective

Even though the present article essentially describes the computer-science basis for a new digital-twin modelling and simulation approach, this work aims to contribute to the broader framework of computational social science (CSS). This designation is certainly at risk of being misunderstood as a branch of social science only. Quite the contrary, however, CSS is an emerging interdisciplinary field of research of growing importance at the intersection of computer science, statistics, and social science. On the one hand, CSS includes a well-known strong formal modelling/simulation branch on artificial societies [1]. On the other hand, it includes the trend towards a dynamically increasing field of automated collection of digital-trace and text data on the Internet that goes hand in hand with an equally important rise of widespread applications of machine learning techniques in society [2]. Though both CSS branches undoubtedly offer valuable scientific insights on their own right, the link between the modelling/simulation branch, on the one hand, and the social-research and social-media research branch [3] respectively, on the other hand, currently remains a quite challenging task still.

### 1.2. Why Agent-Based Models?

This work focuses on agent-based modelling (ABM) of social interaction as well as socio-technical systems. The main advantage of ABM over analytical or machine learning methods is its implementation of self-organisation using simple behaviour and interaction models. Typical analytical methods that can be combined with agent-based modelling are pattern recognition and cluster graph analysis [4], but the required functional social models on a global interaction scope are not always available. Common social models oversimplify the real world, a disadvantage that can be avoided by agent-based simulation only using neighbouring interaction. Agent-based models are especially relevant to simulating social phenomena that are inherently complex and dynamic [5]. Dealing with complexity is a challenge in social modelling, and the decomposition of complex systems in many simple interacting systems (divide-and-conquer approach) is well established and reflected by agents. The simplest agent model only consists of condition–action rules, but is powerful enough to model traffic and crowd flows.

### 1.3. Why Not Analytical Models?

ABM is in principle the counterpart to analytical modelling and analysis. ABM addresses local interaction between agents posing emergence (the global behaviour), whereas analytical models often cover the mapping of individual behaviour on global behaviour directly (the hard-to-predict aggregate outcome) [6]. Analytical models are well established in social science and address both empirical and formal methods that pose their strength with respect to some of the fundamental analysis challenges in social science [7]. The main issue of analytical modelling, e.g., spatial analysis, is their limitation when it comes to covering a broad diversity in social models, i.e., addressing minor variances that occur in real-world systems [8]. This work demonstrates the introduction of the variance of real humans by crowd sensing integrated into the agent-based simulation. It could be shown that small local disturbances have a significant effect on global structures and that interaction is not covered by current analytical models. But ABM can rely on analytical micro-scale models, shown, e.g., by the Sakoda segregation model and the social expectation function (see Section 7).

### 1.4. Explaining Simulation-Based Aggregate Effects

This Section concerns, for instance, the ways and possible improvements of empirically testing assumptions and predictions of such simulation models by experiments and survey research, as detailed e.g., in [9,10]. As outlined there, the validation of empirically grounded agent-based models certainly has to consider the targeted degree of realism and the related importance attached, particularly to mechanism-based explanations. This means, notably, the validation of the mechanisms assumed to produce the expected aggregate effects in the micro-to-macro transition, and in doing so, the rejection of the popular as-if attitude in model-based explanations [6]. Adoption of this approach essentially implies the view that validation should not be restricted to empirical tests of the observable model implications alone, simply because different models can imply the same implications; validation should also extend to the very model assumptions from which the observable implications were deduced.

### 1.5. Why Humans-In-The-Loop?

Such assumptions are essentially assumptions about the behaviour of individual agents and why they act as they act. From a sociological point of view, this brings the humans in the loop and the factors underlying their behaviour (e.g., their preferences, expectations, values, attitudes, personality traits, habits, and resources). It also introduces the social relations that agents maintain in dyads, networks, and larger groups. This, in fact, is important for an understanding of the intended and unintended, even paradoxical emergent effects which result from the behaviour of individual agents and their interactions in society. This, moreover, is important for intervention and steerage purposes too. Cases in point of relevant effects certainly include shapes of structural differentiation (segregation vs. intersection) and opinion formation, currently with strong scientific attention to opinion polarisation (e.g., [11,12]) and extremism due to propaganda in digital social networks [13].

Among considering humans as sensors feeding the simulation at run-time, the loop considers humans as actors, too. i.e., the loop provides output data for crowd and flow control, human decision making via social media, or at least influencing the real world with data from simulations.

### 1.6. Why Simulations in Real-Time?

Social simulation using mathematical, statistical, and agent-based models is well established to investigate and predict social and socio-technical interaction. Often there is a gap between simplified modelling and real-world observations. The factors underlying human behaviour can change over time. Some tend to volatility, some to persistence instead, in any case these factors represent genuine sources of variation. Given that a relevant factor tends to create changes in shorter rather than longer periods of time, and given that such a factor has a share in producing an aggregate effect, models which are capable of both sensing such changes and simulating the resulting effects in real-time appear most suitable for the testing of relevant assumptions of simulation models and the possibly wanted derivation of policy recommendations. Most notably, especially in simulations that are capable of dynamically adapting to changing preconditions at the agent level, afford the opportunity to develop realistic scenarios in use-fields where the level of behavioural change per unit of time is rather high than low.

Crowd simulation can be utilised to feedback data from the simulation to the real-world to control crowd flows, e.g., in cities or domestic services. But this feedback requires the real-time and time-lapse capability of the simulation, discussed in Section 4. The de-facto standard in traditional social simulation is the *Netlogo* simulator [14]. Integrating data mining in agent-based modelling and simulation was first introduced in [15], but data mining uses real-world data collected prior to simulation (delayed coupling of real and virtual worlds).

### 1.7. Why Sensing?

A real-time human-in-the-loop simulation needs continuously updated empirical information on the relevant model parameters. The sensing and collecting of continuous measurements represent the first hub, and the integration of related subjective and objective measurement a second one. It then simply depends on the angle of view which type of measurement is regarded as the primary source and which one is the possible enrichment. In the same way as one can enrich survey data with objective sensor measurements [16], one can enrich sensed physical data with information usually obtained through survey research. One can even discard the weighting inherent to highlighting one source as primary, and just talk of empirical (objective and subjective) information obtained by sensing. In this spirit a key concept of this work is the consideration of humans as sensors and mobile devices such as smartphones are the instruments for sensing the required empirical information. Data fusion approaches can further enhance analytical as well simulative methods [4].

### 1.8. Why Crowd Sensing?

In order to investigate complex and dynamical societies, appropriate data are required. Unfortunately, acquiring such data is a challenge. The traditional methods of analysis in sociology gather qualitative data from interviews, observation, or from documents and records, and carry out surveys of samples of people [5].

The crowd comes in for three related reasons. First, the nearly ubiquitous use of mobile devices makes it simply possible to sense the required information on a large scale, at least in principle. Secondly, a crowd is necessary: without the information obtained from the set of persons that constitutes a crowd, no simulation would be possible at all. Thirdly, the focus on crowds goes along with two ongoing paradigm shifts in social research. The first of these paradigm shifts concerns the changing survey landscape (trend towards the use of non-probability samples) and the second one concerns the tendency from survey research towards social media research. Moreover, just recently, social research encountered the suggestion towards the creation of mass collaboration for data collecting purposes [17]—a trend in line with similar suggestions towards citizen sciences.

The analysis of crowd sensing data and crowd behaviour can be performed with different methods: 1. analytical methods [4]; 2. machine learning (ML); 3. simulation. Simulation and ML can be seen as counterparts to analytical methods or can be used in conjunction.

One major drawback of classical surveys is that they come from measurements made at one moment in time [5]. Crowd sensing combined with real-time simulation can extend the data time scale significantly.

### 1.9. The Role of Incentive Mechanisms

Crowd sensing can be participatory or opportunistic. Furthermore, crowd sensing can be classified in platform- or source- and user-centric architectures [18]. In all cases incentive mechanisms have a high impact on crowd user selection and participation, directly affecting the quality of sensed data [19]. Although the usage of mobile phones introduces new opportunities in social data sensing, adapted incentive mechanisms are required. Distributed and self-organising crowd sensing commonly requires the installation of software, a high barrier for most users, with an impact on participation and data bias (young users vs. older users). Mobile chat-bot agents, e.g., embedded as (*JavaScript*) code in a WEB page (but executed on the user device) as considered in this work can overcome this limitation.

Among participation, the quality of the sensed data contributed by individual users varies significantly by crowd sensing [19]. Sensor fusion and localised pre-processing (filtering) are required. Correlating the quality of information provided by user to the incentive rewarded and as part of the auction model can improve overall data quality of mobile crowd sensing significantly.

### 1.10. Smart City Management

Today administration and management of public services and infrastructure relies more and more on the user and ubiquitous data collected by many domestic and private devices including smartphones and Internet services. People use social digital media extensively and provide private data, enabling profiling and tracing. User data and user decision making have a large impact on public decision-making processes, for example, plan-based traffic flow control. Furthermore, intelligent behaviour, i.e., cognitive, knowledge-based, adaptive, and self-organising behaviour based on learning, emerges rapidly in today’s machines and environments. Social science itself exerts influence on public opinion and decision formation.

Smart city infrastructures and management are characterised by adaptive and self-organising control algorithms using real-world sensor data. In [20], crowds are considered a collective intelligence that can be employed in smart cities, although such self-organising and self-adapting intelligence driven by a broad range of individual goals is inaccurate and can compromise society’s goals.

Traffic is a classic example of group decision making with (social) neighbouring interaction, but commonly, traffic control relies on physical sensor processing only [21]. In [21], adaptive and partly self-organising traffic management was achieved by using agents with multi-levels of decision making and a hierarchical organisational structure. Car traffic control in smart cities can be achieved by global traffic sign synchronisation. But pedestrian, bicycle, and domestic traffic usage cannot be controlled this way. A future vision to control domestic traffic flows is the deployment of digital and social media with chat bots to influence people’s decision making regarding goal-driven mobility, sketched with the methods introduced in this paper.

Traffic control can be performed by perception and analysis of vehicle and/or crowd flows. Furthermore, vehicle-flows can be classified, e.g., introducing weights for individual and public vehicles.

Surveys play another important role in the modelling and understanding of interaction patterns. Artificial Intelligence (AI) and chat bots shift classical survey methods towards computational methods introducing possible implications, i.e., usage of different data, different methods, exposition to different threats to data quality (concerning sampling, selection effects, measurement effects, and data analysis), and different rules of inference are likely to result in comparably different conclusions and public reports, and hence in different input to public opinion formation. Among field studies, simulations can contribute to the investigation and understanding of such interactions.

### 1.11. The Underlying Computer-Science Perspective: A New Augmented Simulation Paradigm

In real worlds, hardware and software robots can be considered close together in a generalised way by an association with the agent model. One prominent example of a software robot is a chat or social bot. Additionally, real and artificial humans can be represented by the agent model, too. Multi-agent systems enable modelling of agent interaction and emergence behaviour. Agent-based modelling and simulation is a suitable methodology to study interaction and mobility behaviour of large groups of relevant human actors, hardware, and software robots, using methods with inherent elements of AI (e.g., learning algorithms) or which use data which may be influenced partly by bots.

Agent-based methods are established for the modelling and studying of complex dynamic systems and for implementing distributed intelligent systems, e.g., in traffic and transportation control (see [21,22]). Therefore, agent-based methods can be described by the following taxonomy [15]:Agent-based Modelling (ABM) - Modelling of complex dynamic systems by using the agent behaviour and interaction model ⇒ **Physical agents**Agent-based Computing (ABC) - Distributed and parallel computing using mobile agents related to mobile software processes ⇒ **Computational agents**Agent-based Simulation (ABS) - Simulation of agents or using agents for simulationAgent-based Modelling and Simulation (ABMS)Agent-based Computation, Modelling, and Simulation (ABX) - **Combining physical and computational agents**

The fifth paradigm is the novelty introduced in this work with the application to social mobility and crowd interaction simulation.

Two promising fields of current studies in computer science are data mining (DM) and Agent-based modelling and simulation (ABMS) [15]. An agent-based simulation is suitable for modelling complex social systems with respect to interaction between individual entities, manipulation of the world, spatial movement, and emergence effects of groups of entities. The main advantage is the bottom-up modelling approach composing large-scale complex systems by simple entity models. The main disadvantage of ABM is the (over-) simplified entity behaviour and simplification of the world the entities are acting in. Commonly, simulations are based on synthetic data or data retrieved by field studies. Many simulations and models are lacking in diversity that exists in the real world. Commonly, sensor and model data (parameters) used in simulations (virtual world) are retrieved from experiments or field studies (real world), shown in Figure 1.

But there is neither feedback from the virtual to the real world nor an interaction of the real world with the virtual world. For example, in [21] self-organised traffic control was simulated with agents by using the JACK software for intelligent agents, which is an agent-oriented development environment built on top of and integrated with the Java programming language and Matlab software. The simulation was based on a parameterised traffic model. Sensor data were created synthetically within the simulation without real-world coupling.

Parameterised models (modelling behaviour and interaction), discussed in Section 3, play an important role in ABM and ABS. The achieved results and conclusions rely on the parameter settings, and the selection of appropriate and representative parameter sets is crucial for modelling real-world scenarios accurately. Furthermore, parameter sets derived from real-world sensing enables variations and evaluation for different real-world situations.


*To overcome the limitations of closed and synthetic simulation worlds, we investigated a new augmented agent-based simulation paradigm coupling real and virtual worlds in real-time, although this is an optional feature.*


Controlling ABS by human participation appears in early work in stock market analysis and prediction [23].

ABS and ABMS are commonly constructed using collections of condition–action rules to be able to percept and react to their situation, to pursue the goals they are given, and to interact with other agents, for example, by sending them messages [5]. The *NetLogo* simulator is an example of an established ABS tool used in social and natural sciences [14], but is limited to behavioural simulation only (ABM domain). In this work, a different simulation approach combining ABM, ABC, and ABS methodologies, is used by deploying the widely used programming language JavaScript. JavaScript is increasingly used as a generic programming language for ABC and ABS [24,25].

### 1.12. Sensing the World

In crowd sensing, users of mobile devices are sensors and actuators [20]. Mobile devices like smartphones are valuable sources for social data [26], either by participatory crowd sensing with explicit participation of users providing first-class data (e.g., performing surveys or polls) or implicitly by opportunistic crowd sensing collecting secondary class data, i.e., traces of device sensor data delivering, e.g., actual position, ambient conditions, network connectivity, and digital media interaction. Crowd sensing and social data mining as a data source contribute more and more to investigations of digital traces in large-scale machine–human environments characterised by complex interactions and causalities between perception and action (decision making). This is relevant for the algorithmic and architectural design of smart cities [20]. Agent-based software is already used in crowd sensing applications [27], which has advantages in adaptivity over traditional static client–server architectures.

But mobile devices are not limited to being sensors. They can be used as actuators, too, by interacting with the user via the chat dialogue or social media that can affect the user behaviour (e.g., mobility decisions).

It is difficult to study such large-scale data collection, data mining, and their effect on societies, domestic services, and social interaction in field studies due to a lack of reliable data and complexity. Controlling all aspects of the virtual world allows explaining the source of structural variation in human interactions, i.e., tackle the hard-to-observe mechanisms of micro–macro interactions. For instance, it is known that the creation processes of social networks rely heavily on the (organisational) environment [28]. Utilising real-world interactions together with ABMs in carefully selected, systematically diverging environments creates a framework in which we can experimentally distinguish between ideal typical situations and understand how different structures result in similar/diverging behaviour. First applications might involve hierarchical/unstratified environments represented by different types of organisations. In so doing, we would add to the question of how the same micro-mechanisms (e.g., homophily, balance) are created by and reproduce different network structures and, hence, extend emerging approaches on network ecological theory.

Agent-based modelling of socio-technical systems is well established [29] and can be applied for smart city management, however commonly applied in an artificial world, i.e., a simulation is performed in virtual reality worlds only to derive and proof models under hard limitations. In this work, a new concept and framework for augmented virtual reality simulation are introduced, suitably, but not limited to, investigate large-scale socio-technical systems. Mobile agents are used already successfully in field crowd sensing [30]. In this work, mobile agents are used to combine in-field ubiquitous crowd sensing, e.g., performed by mobile devices with simulation.

A chat bot agent is capable of performing dynamic dialogues with humans to get empirical data into the simulation and to propagate synthetic data from the simulation world into the real world. The chat bot can act as an avatar providing information for users, e.g., for optimised and dynamic traffic control based on real (covering actual and history data) and simulated data (addressing future predictions).

Chat bot agents (as computational agents) can operate both in real and virtual worlds including games and providing a fusion of both worlds by seamless migration. Mobile computational agents, i.e., mobile software, have advantages in strong heterogeneous systems, existing in opportunistic and ad-hoc crowd sensing environments. Examples are traffic control by smartphone data. Agents are loosely coupled to their environment and platform and interact with each other, e.g., via tuple spaces (generative data-driven communication) and via unicast or broadcast signals (addressed messages) [31].

The novel MAS crowd sensing and simulation framework, introduced in the next sections, is suitable to combine social and computational simulations with real-world interaction at run-time and in real-time, e.g., by integrating crowd sensing, using mobile agents. There are two classes of mobility that can be modelled in the simulation: (1) Social mobility on short- and long-term time scales, e.g., creating group formation and segregation, and (2) goal-driven mobility, e.g., creating traffic.

Agent-based traffic management simulation still neglects social interactions and the influence of social media, e.g., MATISSE, the Multi-Agent based Traffic Safety Simulation System [32], which is a large-scale multi-agent-based simulation platform designed to specify and execute simulation models for agent-based intelligent transportation systems.

Among the capability of real-time simulation, the simulation framework is capable of creating simulation snapshot copies that can be simulated in parallel in non-real-time to get future predictions (simulation branches) from actual simulation states that can be backpropagated into the current real-time simulation world.

## 2. Intelligent Machines and Socio-Technical Systems

There are basically three classes of entities existing in real and virtual worlds: 1. Humans, 2. machines, and 3. robots. Automatic and autonomous vehicles belong to class 3; semi-automatic driving is, e.g., relevant for smart traffic flow management.

Human–machine interaction is an emerging field. The interaction with intelligent machines is much more sophisticated and created two classes of interaction: 1. human–machine (HM), and 2. machine–machine (MM). Today, robots can be considered as intelligent machines (i.e., AI bots) posing some kind of autonomous behaviour, learning, and interacting with environments and humans. The main advantage of ABMS and the proposed agent-based framework is the unified handling of totally different entities using the same agent model. Autonomous vehicles are examples of both HM and MM interaction.

There is a micro-context of everyday life and the macro-context of public decision and opinion formation related to human–robot interaction, affecting, e.g., future city management like adaptive and dynamic traffic control considering a broad range of participants and vehicles (cars, buses, trains, pedestrians, and bicycles). Automatic and probably autonomous driving systems can be considered as intelligent machines (i.e., robots) interacting with humans and environments, too. Social behaviour of humans and interaction with intelligent machines will depend on the way intelligent machines will be controlled and the way they interact with humans and other intelligent machines. For example, chat bots influence human opinions, but humans and other chat bots can influence the behaviour and knowledge base of chat bots, too. There are different involved ways of mutual affection that are difficult to investigate in classical field studies. The proposed simulation method integrating real-world environments by using mobile agents can boost investigations of such complex interacting environments. For example, decisions of traffic participants (e.g., car drivers) cause wide-ranging effects on global behaviour (emergence), and centralised or decentralised traffic control will influence individual behaviour.

Since both sides (humans and bots) may learn from each other in this interaction, the studies shall explore possibilities of targeted responses of the human side. That way we expect to reveal options of understanding the influence on the possible ways that AI will shape future society. In addition, we expect to find out the factors in robot–human interaction that let AI gain acceptance in society. This list of factors shall, amongst others, include the visibility of AI during a communication (e.g., can a human discern whether a comment comes from a bot or a human and what difference this makes to his/her response?) and different programmable behavioural and ethical principles in robot–human interaction (by systematically varying the reward function). Furthermore, since we expect in the medium- and long-term, respectively, a trend towards the use of robots that not only assists but replace humans in daily communication, we project the development and exploration of the first occurrence of this in a home environment, and the evaluation of its applicability in practice.

At society’s macro-level, robot–human interaction takes place too, though less visible and less direct. The investigation of public decision and opinion formation is of high relevance for the design of public management services. We project an exploration of how this process is shaped in the interaction of groups of relevant human actors who interact with intelligent machines through its reports of findings of projects which either uses methods with inherent elements of AI (e.g., learning algorithms) or which use data which may be influenced partly by bots. In this regard, the study is targeted on possible emergent effects of AI at the system level of society.

To summarise, the agent model used in this work captures different entity classes, mapping activities (performing actions like environmental interaction) *A*, transitions *T* between activities (that can be conditional based on *V* and *S*), data variables *V*. and sensors *S*, on actions *X*, discussed in Section 4:(1)Ag(A,T,V,S):S×A×T×V→X.

## 3. Parameterised Modelling and Emergence

Modelling of social and socio-technical systems can be classified in macro-scale and micro-scale models addressing a macro- and micro-context discussed in the previous section.

Commonly, social behaviour is studied on a macro-scale level with a unified model assuming a common mean of individual behaviour or on a micro-scale level with a unified behaviour model. Social behaviour and interaction models with parameters are used to study more heterogeneous and individual behaviour beyond a unified mean.

On a micro-scale level, it is assumed that there is an individual parameterised behaviour model *M* for individual entities or groups of entities, i.e., a model of social interaction, which is used to model physical entities and digital twins representing individual real humans (with *P*: Parameter set, *S*: Sensor set, *X*: Action set):(2)M(S,P):S×P→X.

A set of individual entities (agents) modelled by *M*(*P*) interacting with each other within a bound space will create a specific observation on system level, i.e., emergence effects. One example is a traffic jam as a result of individual behaviour and constraints.

## 4. Simulation Architecture and Agent Platform

The proposed framework couples virtual and real worlds by integrating simulations with human interactions by using two classes of agents: (1) Computational agents (chat bots), and (2) physical agents, both executed inside the simulation world by a unified agent processing platform. The agent-based simulation assumes a spatial context and agent interaction within a spatial domain (or range), similar to cellular automata.

The entire crowd sensing and simulation architecture consists of the following components:Unified **Agent Processing Platform** programmed in *JavaScript* for high portability and deployment in strong heterogeneous environments: JavaScript agent machine (JAM) [31];**Crowd Sensing Software** (Mobile App and WEB Browser using JAM);Agent-based simulation with Internet connectivity supporting three different agent types:○**Physical behavioural agents** representing physical entities, e.g., individual artificial humans;○**Computational agents** representing mobile software, i.e., used for distributed data processing and digital communication, and implementing chat bots;○**Simulation agents** controlling the simulation and performing simulation analysis (e.g., creation of physical and computational agents, reading and writing sensor data, accessing databases)Chat dialogues, chat bots, and mobile agents collecting user and device sensor data;**Knowledge-Based Question-Answer Systems, Natural Language Processing**.

All agents are programmed in *JavaScript* and executed by the JAM platform in a sandbox environment. Computational agents can migrate between platform nodes (as mobile process snapshots). This widely-used programming language offers a steep learning curve and *JavaScript* can be executed on a wide range of host platforms and software. This feature includes the execution of mobile agents with a WEB browser by embedding the agent platform in WEB pages. *JavaScript* programs and agents offer high portability which is essential for the deployment in strong heterogeneous environments and to reach a broad range of crowdsensing data providers.

Simulation is performed by the Simulation Environment for *JAM* (*SEJAM*, details are discussed in [31,33]). In this work, the *SEJAM* simulator is used to create a simulation world (consisting of entities represented by agents) that is attached to the Internet enabling remote crowd sensing with mobile computational agents. The *SEJAM* simulator is basically a graphical user interface (GUI) on the top of *JAM*, shown in Figure 2.

*SEJAM* extends *JAM* with a simulation layer providing virtualisation, visualisation of 2D- and 3D-worlds, chat dialogues and avatars, and an advanced simulation control with a wide range of integrated analysis and inspection tools to investigate the run-time behaviour of distributed systems. A *SEJAM* world consists of one physical *JAM* node and an arbitrary number of virtual/logical *JAM* nodes associated with a visual object (shape), which can be connected via arbitrary connection graphs to establish communication and migration of computational agents. In the graphical 2D world virtual *JAM* nodes can be either placed at any coordinate or aligned on a grid world.

Each *JAM* node is capable of processing thousands of agents concurrently. *JAM* and *SEJAM* nodes can be connected in clusters and on the Internet (of Things) enabling large-scale "real-world"-in-the-loop simulations with millions of agents. Agents executed in the simulation have access to an extended simulation API. Physical agents can use an extended *NetLogo* compatible simulation API extension, too, enabling simulation world analysis and control, based on object iterators (*ask* and *create* statements, see [14], with a detailed discussion in Appendix A).

The mixed-model simulation world consists of physical and computational agents bound to logical (virtual) platforms (host of the agent) that are arranged or located on a lattice (patch grid world) to provide world discretisation for the sake of simplicity. The patch grid world was derived from the *NetLogo* simulator, although the *SEJAM* simulator is not limited to this world model and can handle arbitrary two-dimensional non-discretised world coordinate systems (e.g., GPS). The agents are mobile. Computational agents, as mobile software processes, can migrate between platforms (both in virtual and real digital worlds), whereas physical agents are fixed to their platform and only the platform is mobile (in the virtual world only).

The *JAM* agent behaviour model is based on activity-transition graphs (ATG, details in [34]). An ATG decomposes the agent behaviour into activities performing actions (computation, mobility, and interaction with other agents and the world). An activity is related to a sub-goal of a set of goals of the agent. There are transitions between activities based on the internal state of the agent, basically related to reasoning behaviour under both pro-active (goal-directed) and reactive (event-driven) stimuli. Modification of agents (visual, shape, position, body variables, and state) can be performed globally by the world agent via a simulation interface providing a *NetLogo* API compatibility layer, or by the individual agents.

Both *JAM* and *SEJAM* are programmed entirely in *JavaScript*, enabling the deployment on a wide range of host platforms (mobile devices, servers, IoT devices, and WEB browser, details described in [35]). *JAM* and *SEJAM* can be connected via IP-based communication links. *JAM* provides virtualisation and security (encapsulation) by the agent input–output system (AIOS), tuple spaces for generative or signal-driven inter-agent communication, and virtual (logical) nodes bound to a world contained in one physical *JAM* node. Each physical or logical *JAM* node can be connected with an unlimited number of remote *JAM* nodes by physical links (UDP/TCP/HTTP using the AMP protocol), shown in Figure 2. Logical nodes can be connected by virtual communication links. Links provide agent process migration, signal (message) and tuple propagation.

The agent behaviour defines:a set of sensors and beliefs about the world (its data set), basically stored in the agent’s body variables;a set of events that it will respond to (signals, tuples, sensor changes);a set of goals that it may desire to achieve, basically implemented by the set of activities, and;a set of plans that describe how it can handle the goals or events that may arise activating activities.

A simulation world consists of multiple virtual *JAM* nodes (*vJAM*), which can be connected by virtual links. In contrast to pure ABM simulators like *Netlogo*, the *SEJAM* simulator simulates computational and physical agents and the agent processing platform itself and is primarily an ABC simulator, but can be used for ABM like simulations, too. There is a global simulation model defining agent classes (behaviour and visuals), nodes (visuals), resources, simulation parameters, and the construction of the simulation world.

Computational agents are always bound to a virtual *JAM* node. A node can be created dynamically at simulation run-time or during the initialisation of the simulation world. A node is related to a virtual position in the two-dimensional world (similar to a patch in the *Netlogo* world model). Virtual nodes can be grouped in tree structures and the movement of a parent node (in the virtual world) moves all children, too.

In contrast to a *Netlogo* simulation model using one main script describing the entire world, agents, and resources, a dedicated world agent (operating on a dedicated world node) controls the simulation in *SEJAM*. A node position can be changed by agents (related to the movement of turtles in the *Netlogo* model. To summarise, *JAM* agents play two different roles in the simulation: (1) ABC: mobile crowd sensing, digital interaction between physical agents and the environment, and simulation control and; (2) ABM: Representation of humans, bots, and digital twins of humans.

There is a significant difference between traditional closed-world simulations and simulations coupled with the real world and real-time environments (human-in-the-loop simulation). Closed simulations are performed on a short time-scale with pre-selected use-cases and input data. The simulation can be processed step-wise without a relation to a physical clock. In contrast, open-loop simulation requires continuous simulation on a large time-scale creating big data volumes. A relation to a physical clock is required, too.

The mapping of the physical onto virtual worlds and vice versa is another issue to be handled. There are basically three possible scenarios and world mapping models (illustrated in Figure 3):The real and virtual worlds are isolated (no agent/human of one world knows from the existence of the other);The real world is mapped on the virtual simulation world (simulating the real world with real and artificial humans or entities);The virtual world extends the real world, e.g., by a game world, and real and virtual entities known from each other.

### Simulation Snapshots

One key capability of the simulator and the agent platform is the creation of agent, world, and entire simulation snapshots. This feature enables the forking of simulation runs either continuing on a different time scale (speed-up) or with different parameters and sets of agents. A time-lapsed simulation can be used to get future time predictions, e.g., of crowd or traffic flows, clusters, or social networks.

## 5. Workflow

The following Figure 4 shows the principle work and data flow of the proposed architecture. The ABMS relies on generated data stored in a database provided by synthetic data, Monte Carlo simulation, and data mining (DM) of sensor data from mobile crowd sensing (MCWS). The MCWS is performed by mobile chat bot agents, discussed in Section 6.3, which establishes the connection between virtual and real worlds. But the data flow is bidirectional, and agents can carry data generated by the simulation to mobile devices and users in the real world, e.g., by posting messages in chat blogs or other social media.

Simulation data is stored in the database, too. The simulation data, e.g., monitoring data or artificial sensor data, can be analysed numerically or statistically by DM. Sensor data can be collected off-line (classical surveys) before the simulation or on-line during the simulation creating incremental simulation runs.

## 6. Crowd Sensing and Simulation

### 6.1. Crowd Sensing, Surveys, and Cloning of Digital Twins

The basic crowd sensing and social survey architectures can be classified in (see Figure 5):Centralised crowd sensing architecture with a single master instance (crowd sourcer);Decentralised crowd sensing architecture with multiple master crowd sourcer instances;Self-organising and ad-hoc crowd sensing architecture without any dedicated crowd sourcer master instances.

The survey used in this work relates to the classical central approach, although it can be extended with a decentralised and self-organised approach, too, handled by the same framework and mobile chat bot agents.

Among the crowd sensing architecture, there are two different crowd sensing models:Participatory crowd sensing negotiated between a crowd sourcer and user based on utility and reward; there is strong coupling between crowd sourcer and crowd sensing participants;Opportunistic or ad-hoc crowd sensing with weak coupling between crowd sourcer and participant based on immediate reward.

Classical surveys are participatory crowd sensing and the selection of data provider (user) is done carefully to avoid biased data. Ubiquitous computing enables opportunistic crowd sensing using mobile devices, e.g., smartphones, from a broader range of participants (usually not known) in real-time.

Initially, an AB simulation is performed with artificial agents relying on model parameters derived from theoretical considerations, prior sampled experimental data, and survey data, which can be combined with Monte Carlo methods. Augmented virtuality enables dynamic simulations with agents representing real humans (or crowds). By using crowd sensing it is possible to create digital twins of real humans based on a parameterised behaviour and interaction model, including spatial context. The parameters of artificial humans in the simulation represented by agents are collected by sensor data, i.e., surveys optionally fused with physical sensors like GPS. One simple example is shown in Section 7.

The crowd sensing via mobile chat bot agents enables the interaction of real humans with agents and digital twins in the simulation world in real-time and vice versa. The digital twins, as well as the artificial physical agents in the simulation, can interact by dynamically creating (influenced) dialogues reflecting the state of the simulation world.

The survey performed by chat bot agents (computational agents) aims to create digital twins in the virtual world from survey participants in the real world by deriving twin behaviour model parameters *P* from the survey feedback answers *F* retrieved by a user dialogue *D*. The survey is performed by chat bot agents that can execute dynamic dialogues via a chat dialogue platform based on previous answers and context.
(3)$$survey(D):D\to F,analyse(F):R\to P,twin(P):P\to AG_{twin}$$

Crowd sensing combined with ABM/ABS introduces randomisation and variance in the simulation and model verification, shown in Figure 6. The addition of digital twins with behaviour parameters derived from real humans and CWS can lead to different outcomes of the simulation:The expected structures (emergence) are observed as given by the model (same case as the pure synthetic world simulation);No structures are observed caused by the disturbance of the real-world interaction;Expected structures and new structures are observed normally not occurring in pure synthetic and closed worlds.

The ABS/CWS coupling via digital twins enables iterative surveys in multiple rounds, i.e., results from one round combined with new simulation results can be used for the next survey round.

One major issue in simulation and experimental field studies is the reproducibility of results. To ensure reproducibility and to avoid data bias, participants of surveys are commonly chosen carefully and not randomly (participators crowd sensing). Opportunistic crowd sensing selects users randomly and requires further data analysis and user classification to ensure simulation stability.

### 6.2. Security

Participatory crowd sensing has to ensure data privacy and protection against data manipulation. This crowd sensing strategy can be established via secure channels and by using client–server architectures. Opportunistic crowd sensing, on the other hand, poses a loose coupling of data producers and consumers, raising security issues.

Security is provided on different levels, either by the *JAM* platform, e.g., capability protected agent roles, or by system-level agents, e.g., the chat mediator agent, which authorises and filters survey requests, or detects spam or malware agents. A crowd sensing platform can be open (like any other WEB page), or protected via a capability.

#### 6.2.1. Agent Roles

Physical agents are caught in the simulation world. They can access and modify any other agent in the virtual world and the simulation world itself. They are executed on a privileged level. But computational agents can leave the virtual world and enter the real world and vice versa. The loose coupling of the simulation with the crowd sensing network and to enable opportunistic sensing requires different agent privilege roles. Four different roles are supported by the platform:Guest (not trustful, non-mobile);Normal (maybe trustful, mobile);Privileged (trustful, mobile);System (highly trustful, local processing only, non-mobile, creation by platform only).

The agent platform assigns a security level to new received agents (commonly level 0 or 1). The agents have to negotiate a higher level using the capability-based approach discussed in the next sub-section. The privilege level determines the set of API functions that an agent can access (e.g., migration, forking, and communication).

#### 6.2.2. Capabilities

Capabilities are keys to enable specific agent API functions like migration. A capability consists of a service port, a rights bit field, and an encrypted protection field generated with a random port known by the server (node) only and the rights field. The rights field enables specific rights. e.g., the right to negotiate a higher privilege level. A capability is created by the respective service (e.g., the platform itself) by encoding the rights field in a secure port using a private port and a one-way function, shown in Figure 7. Major issues are the secret handling of capabilities on agent migration (requiring secured channels) and passing the right capability to an agent. A broker server is commonly required to distribute and pass capabilities to agents based on classical authorisation and authentication.

Capabilities enable fine-grained control of operations that can be performed by agents and allow loosely coupled self-organising systems, e.g., participation in chats. In self-organising systems, the identity of agents is not of primary interest. Instead the actions they can perform are of primary interest.

### 6.3. Chat Bots as Mobile Agents in Both Worlds

Mobile agents are used in this work for distributed data processing and data mining in the real and virtual simulation world seamlessly. It is assumed that the real-world environments consist of communication access points (i.e., beacons, e.g., using WLAN and WWAN communication technologies) and mobile devices (smartphones, IoT devices). Commonly there are no globally visible organisational and network structure, i.e., agents cannot rely on a network world model to reach and identify specific devices. Each *JAM* node provides connectivity information, i.e., a list of all connected *JAM* nodes. Commonly, *JAM* nodes are connected peer-to-peer in mesh-like networks via TCP/HTTP or UDP connections.

Agents can communicate with each other either by exchanging tuples via a tuple space database or by using signals (lightweight messages) that can be propagated remotely along agent migration paths. Tuple space access is generative communication, i.e., the lifetime of tuples can exceed the lifetime of the generating agent. Additionally, tuple space communication is data-driven and anonymously (sender and receiver need no knowledge about each other).

Mobile agents, e.g., chat bots, can be used to carry information from one location to another. Moreover, mobile devices carrying mobile agents can be used to collect, carry, and distribute data within large regions via tuple spaces, shown in principle in Figure 8. The virtual simulation world is just another region in the mobility regions of agents, and chat bots performing crowd sensing can migrate between real and virtual worlds seamlessly.

### 6.4. Chat Bots and Human–Agent Interaction

Mobile agents can migrate between different devices. The main goal of explorer agents is to collect crowd and social data. This data can be retrieved by device sensors (including aggregated virtual sensors) and information provided by users. Privacy and security have to be addressed. Among device sensor data like position, velocity, and ambient light, which can be collected by mobile explorer agents automatically, user data can be gained by a human–agent dialogue chat via a WEB Browser or Android/iOS App (see Figure 9). Therefore, an explorer agent is capable of interacting with humans by performing textual dialogues (question–answer surveys). Due to the distributed and parallel processing of loosely coupled agents filtering and scheduling of agent dialogue request has to be performed. On each device participating in crowd sensing there is a mediator agent that manages and assesses interaction requests from incoming (explorer) agents. Passed requests are forwarded to a user–bot chat dialogue. Responses to questions (requests) are passed back to the requesting explorer agents.

Each agent migrating to a device operated by humans (i.e., smartphones, navigation computer, or fixed terminals) can ask questions via the APP tuple space by inserting a question tuple, which is handled by the node mediator agent. The tuple space enables data-driven communication between agents that have different roots, a situation arising in opportunistic crowd sensing. See [35] for details on tuple-space communication.

A question request tuple is evaluated by the mediator agent performing security and privacy checks, finally passing the question to the chat blog waiting for human participation. The answer (if any within a defined timeout) is passed back to the original requesting agent by an answer tuple, again previously checking privacy and security concerns.

The principle communication format via the tuple space is shown in Algorithm 1. The dialogues can be composed of dialogue trees, i.e., questions and messages that can be dynamic and can depend on previously answered questions or environmental perception (sensors).

**Algorithm 1:** Communication of remote chat bot agents with user dialogue mediator agents via an anonymous tuple space1out([‘Message’,<id>,<message>])2out([’Question’,<id>,<question>,3{value: <default>, 4     type:’number’|‘string’, 5     icon:<icon-name>}]6  inp.try(timeout, 7    [’Answer’,<id>,<question>,_],8     **function** (tuple) { 9      **if** (tuple) this.result = tuple[3]})

In doubt, the question or answer is discarded or modified (e.g., filtered or annotated with additional warnings). The mediator agent performs question request scheduling to satisfy agent requests and human interaction capabilities (high question rates will decrease the user interest and motivation to answer questions). Additionally, the mediator agent has to monitor chat bot dialogues by using natural language processing (NLP).

A typical chat dialogue is shown in Figure 9. It consists of question–answer snippets processed and controlled by the mediator agent operating on each device with user interaction. This chat dialogue is embedded in a *JAM* platform application software (*JAMapp*). *JAMapp* is available for smartphones or as a WEB-page that can be processed by any *JavaScript* capable WEB browser.

The mediator agent can pass questions (or just stand-alone messages) to the user chat via a simple platform API providing a question text, possible answers or input fields, and a callback function handling the result, shown in Algorithm 2.

**Algorithm 2.** Chat dialogue API accessed by the mediator agent1chat.message(<id>,<message>);2  chat.question(<id>,<question>,{3    size: <test-size>,4     icon: <icon-name>,5     value: <default value>6     sub_type: ‘number’|‘string’,7     placeholder: <placeholder-value>8  },function (result) {9   this.answer = result;10  },timeout);

The following Definition 1 describes the dialogue script passed to a chat bot agent that is used to perform the crowd sensing on mobile devices by the chat bot agents. The question dialogue is dynamic: (1) Some questions are only presented based on answers of previous questions → *cond* function, and (2) some question texts are updated based on previous answers → *eval* function replacing text variables $ in the question text with values returned by an array. The optional *param* function can be used to change a default value or a choices list (returning an array of alternatives). Among user dialogues, device sensors (e.g., light intensity or position) can be requested within the dialogue script, too.

**Definition** **1.**
*Survey job type definition.*


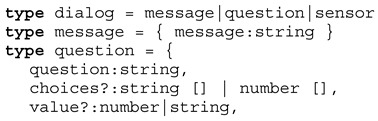



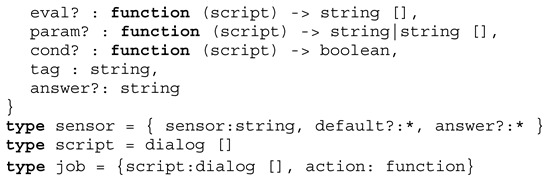



## 7. Social Interaction and Mobility by the Sakoda Model

To demonstrate the augmented virtuality approach combining agent-based simulation with agent-based crowd sensing, social interaction modelled by the Sakoda model [36] was chosen as a simple social interaction and behaviour model between groups of individual humans posing self-organising behaviour (emergence) and structures of social groups by segregation.

The more well-known Shelling model used to study segregation effects (long-range mobility) bases on the less known but more general Sakoda social interaction model. The Sakoda model can be used for long- and short-range mobility, required, e.g., to study interaction and group aggregation effects, e.g., in cities and traffic.

Modelling social interaction is an example of an individual parameterised behaviour model explained below. The parameters can be set a priori or derived by surveys, i.e., using crowd sensing. In this case, the crowd sensing is only used to provide input data for the simulation. There is no immediate feedback to the crowd.

The Sakoda behaviour model can be applied to two different spatial and time scales resulting in different self-organising behaviour:Long-time and long-range scale of mobility addressing classical segregation, e.g., in cities and countries, andShort-time and short-range scale of mobility addressing, e.g., city mobility.

Both time scales are relevant for future city management including traffic management.

The original Sakoda behaviour model [36] consists of social interactions among two groups of individuals evolving in a network according to specific attitudes of attraction, repulsion, and neutrality. An individual evaluates its social expectative at all possible available locations (starting at its current location), preferring originally those near individuals associated with attractive (positive) attitudes and avoiding locations near individuals associated with repulsive (negative) ones. This procedure is repeated randomly among all possible individuals; henceforth Sakoda’s algorithm is iterated repeatedly developing a spatially distributed social system to an organised spatial pattern, although this depends on the parameter set of the model, crowd densities, and individualisation, introduced below.

### 7.1. Interaction Model

For the sake of simplicity, there is a two-dimensional grid world that consists of places at discrete locations (*x*,*y*). An artificial agent occupies one place of the grid. A maximum of one agent can occupy a place. The agents can move on the grid and can change their living position. It is assumed that there are two groups related to the classes *a* and *b* of individuals. The social interaction is characterised by different attitudes [31] of an individual between different and among same groups given by four parameters:(4)$$S=(s_{aa},s_{ab},s_{ba},s_{bb}).$$

The model is not limited to two groups of individuals. The *S* vector can be extended to four groups (or generalised) by the matrix:(5)$$S_{abcd}=\left(\begin{array}{cccc} s_{aa} & s_{ab} & s_{ac} & s_{ad}\\ s_{ba} & s_{bb} & s_{bc} & s_{bd}\\ s_{ca} & s_{cb} & s_{cc} & s_{cd}\\ s_{da} & s_{db} & s_{dc} & s_{dd} \end{array}\right).$$

The world model consists of N places *x*_i_. Each place can be occupied by none or one agent either of group α or β, expressed by the variable *x*_i_ = {0,−1,1}, or generalised *x*_i_ = {0,1,2,3,4,..,n} with *n* groups. The social expectation of an individual *i* at place *x*_i_ is given by:(6)$$f_{i}(x_{i})=\sum_{k=1}^{N}J_{ik}\delta_{s}(x_{i},x_{k}).$$

The parameter *J*_ik_ is a measure of the social distance (equal to one for Moore neighbourhood with a distance of one), decreasing for longer distances. The parameter δ expresses the attitude to a neighbouring place, given by (for the general case of *n* different groups):(7)$$\delta_{s}(x_{i},x_{k})=\{\begin{array}{ll} s_{\alpha \beta }, & \mathrm{if}\;x_{i}\neq 0\;\mathrm{and}\;x_{k}\neq 0\;\mathrm{with}\;\alpha \;=\;x_{i},\beta \;=\;x_{k}\\ 0, & \mathrm{otherwise} \end{array}.$$

Self-evaluation is prevented by omitting the current place (i.e., *k* ≠ *i*). There is a mobility factor *m* giving the probability for a movement.

An individual agent *ag*_i_ of any group α (class from the set of groups) is able to change its position by migrating from an actual place *x*_i_ to another place *x*_m_ if this place is not occupied (*x*_m_ = 0) and if *f*_i_(*x*_m_) > *f*_i_ (*x*_i_) and the current mobility factor *m*_i_ is greater than 0.5. Among social expectation (resulting in segregation), transport and traffic mobility have to be considered by a second goal-driven function *g*(*x*_m_), commonly consisting of a destination potential functions with constraints (e.g., streets). If *g*_i_(*x*_m_) > *f*_i_(*x*_m_) > *f*_i_ (*x*_i_) than the goal-driven mobility is chosen, otherwise the social-driven is chosen.
(8)g:(xi,xm,xk,v,t)→ℝ

The mobility function *g* returns a real value [0,1] that gives the probability (utility measure) to move from the current place *x*_i_ to a neighbouring place *x*_m_ to reach the destination place *x*_k_ with a given velocity *v*. The *g* function records the history of movement. Far distances from the destination increase *g* values. Longer stays at the same place will increase the *g* level with time *t*. Social binding (i.e., group formation) will be preferred over goal-driven mobility.

The computation of the neighbouring social expectation values *f* is opportunistic, i.e., if *f* is computed for a neighbouring node assuming the occupation of this neighbouring place by the agent if the place is free, and the current original place is omitted *x*_i_ for this computation. Any other already occupied places are kept unchanged for the computation of a particular *f* value. From the set of neighbouring places and their particular social and mobility expectations for the specific agent the best place is chosen for migration (if there is a better place than the current with the above condition). In this work, spatial social distances in the range of 1–30 place units are considered.

Originally, the entire world consists of individual agents interacting in the world based on one specific set of attitude parameters *S*. In this work, the model is generalised by assigning individual entities its own set *S* retrieved from real humans by crowd sensing, or at least different configurations of the *S* vector classifying social behaviour among the groups. Furthermore, the set of entities can be extended by humans and bots (intelligent machines) belonging to a group class, too.

Segregation effects inhibit individual movement until a different social situation enables a movement. Transportation mobility triggers movement even if there is no social enabler. This is reflected in the extended Sakoda model by the mobility factor *m* and the goal-driven expectation function *g* that control mobility and overlays social and transportation and traffic mobility. The mobility function *g* includes random walk and diffusion behaviour, too. Constrained mobility is one major extension of the original Sakoda model presented in this work.

### 7.2. Model Parameters and Crowd Sensing

Creation of virtual digital twins is the aim of the crowd sensing. The crowd sensing is performed with chat bot agents. One stationary agent is operating on a user device, e.g., a smartphone, and another mobile agent is responsible for performing a survey (either participatory with a former negotiation or opportunistic ad-hoc). The results of the survey, a set of questions, are used to derive the following simulation model parameters, shown in Definition 2.

**Definition** **2.**
*Sakoda model parameters.*


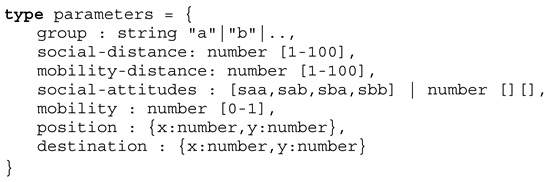



The *group* parameter sorts the user in one of two classes *a*/*b*, the *social-distance* parameter is an estimation of the social interaction distance, the *social-attitudes* parameter is the *S* vector, but limited to a subset of all possible *S* vector combinations (discussed below), and the *mobility* parameter is a probability to migrate from one place to another. The position (in Cartesian coordinates) is derived from the living centre of the user (global position data, GPS) and mapped on the simulation world (*x*,*y*). The *S* vector parameter determines the spatial social organisation structure. Typical examples of the *S* vectors with relation to social behaviour are [36]:(1,−1,−1,1): Typical segregation with strong and isolated group clusters;(0,−1,−1,0): Mutual suspicion;(1,−1,1,−1): Social climbers;(1,1,−1,−1): Social workers;(1,1,1,1): Inclusion;(1,0,−1,1): Traffic (a:cars, b:bicycles).

The crowd sensing extends the simulation with the following dynamics and changes:Enhancement of the synthetic simulation with real-world data by digital twins not conforming to an initial parametrisation of the artificial individuals (affecting *S* vector, spatial social interaction distance, mobility);Adaptation and change of the fraction of different groups (commonly an equally distributed fraction from each group is assumed);Convergence and divergence of the emergent behaviour of group formation (spatial organisation structures);Influence of real-world users by chat bots with data from the virtual simulation world.

The dynamics in the simulation world can be backpropagated to the real world by chat bot agents, too, delivering information and opinions formed by the artificial entities. Among classical surveys, chat bots can perform manipulation by distributing biased information or opinions via the chat dialog interface.

## 8. Traffic Prediction and Control

The previously introduced functional social interaction model is suitable to model group formation, but not group control. Traffic control and prediction of traffic emergence effects, e.g., jams, require a different algorithmic and iterative agent-based model introducing constraints, i.e., traffic signs and signals, security distance, and crash avoidance. In this model, there are vehicles and humans.

Traffic control can be performed by the perception and analysis of vehicle and/or crowd flows. Furthermore, vehicle-flows can be classified, e.g., introducing weights for individual and public vehicles, and crowd-flows can be classified the same way, e.g., distinguishing individuals from public mobility people or leisure from working mobility. In this work, both analysis approaches should be considered to optimize vehicle- and crowd-flows.

Among driver and passenger behaviour estimation by crowd sensing, accurate and robust localisation can be a challenge, especially if it should be performed in a distributed way. Distributed sensor fusion is a key method for deriving data with high quality and strength. Mathematical and statistical methods are well established, e.g., Gaussian processes. In [37], mobile sensors networks and Gaussian Markov random fields are proposed for accurate spatial prediction. The authors could proof the distribution of such method and the deployment in low-resource networks. Clustering effects (e.g., of users and their mobile devices) can have relevant impact on simulation and traffic prediction, too. Gauss–Poisson models as a class of clustered point processes are able to capture such clustering effects [38].

### 8.1. Traffic Model

The traffic model consists of a city map (streets, places, and so on), vehicles, and humans. There are four sub-classes of vehicle agents:An individual vehicle with one driver agent and zero passengers and dynamic route planning;A shared vehicle with one driver and [1,n] passengers and semi-dynamic route planning;A public vehicle with one driver and [1,m] and m >> n passengers and static route planning;Bicycles with one driver.

Each vehicle requires at least one twin agent controlling the vehicle (the driver), although a vehicle (except bicycles) can be driven automatically or autonomously. In the latter case a driver sets the destination position only.

Similar to vehicle agents, there are three different sub-classes of twin agents:A driver agent (can be a passenger, too);A passenger agent using a shared or public vehicle;Pedestrians or currently non-mobile humans.

The crowd sensing provides input data for the simulation by creating parameterised digital twins of real humans, and it provides crowd feedback (output of the simulation), e.g., by suggesting alternative mobility routes or using incentive mechanism to control crowd flows.

### 8.2. Crowd Sensing and Traffic Control

The crowd sensing data can be used to analyse current traffic situations (driver class) and to predict crowd flows resulting in traffic in the near future (passenger and pedestrian classes). The flow, behaviour, and clustering prediction can be fed back to the real world, e.g., by controlling traffic signal switching or by influencing people’s decision making via social or domestic media.

Traffic control can address optimised signal switching as well as dynamic street sign (e.g., velocity), and control of vehicles by influencing drivers (or passengers). Signal control can be static (with respect to switching times) or dynamic, including signal group switching. Commonly signals are controlled by evaluating the current traffic flow situation without evaluating the drivers and passengers (e.g., with respect to destination, attitudes, or goals). If the goal is to optimise the crowd flow rather than the vehicle flow, then public or shared vehicles have to be preferred by priorities.

## 9. Use-Cases and Evaluation

The next sub-sections show three different use-cases showing the deployment of the augmented simulation approach.

The first use-case is related to social science and addresses segregation as well as short-term and short-range mobility arising in city traffic (clustering driven by social interaction and behaviour). It is used primarily to study social networks and interactions. Real and virtual worlds can be different.

The second use-case demonstrates the suitability of the framework for smart light management in cities by using crowd sensing to deliver real-time data from physical sensors and humans. The simulation is used to derive optimal street illumination under different constraints, e.g., low-energy constraints. The simulation is primarily used to control real-world city infrastructure and the real world is mapped on the virtual world.

Finally, the third use-case demonstrates the study of smart traffic control using crowd sensing delivering real-time data from physical sensors and humans. Depending on the goal simulation (control or modelling), the real world can be mapped on the virtual world, but can be different, too, using artificial city maps only.

The first use-case relies on a pure mathematical interaction model, whereas the second and third use-cases rely on pure algorithmic interaction models.

### 9.1. Segregation and Social Interaction

This use-case should demonstrate the impact of real-world humans on the simulation of social interaction and social networks (clusters). Crowd sensing was used to create digital twins in the simulation with model parameters derived by surveys performed with a WEB-page or suing mobile App software. For the sake of simplification, the world consists of a regular grid (patch grid). The Sakoda interaction model introduced in Section 7 is basically a local interaction model, in contrast to algorithmic models shown in the use-cases in Section 9.2 and Section 9.3.

#### 9.1.1. Long-Term Mobility

The initial simulation was performed with 200 class *a* and 200 class *b* agents and a unified *S* = (1,−1,−1,1) setting for all agents (blue and red squares) of both classes and unconstrained mobility leading to classical segregation structures (strong isolated clusters), shown in Figure 10.

During the simulation run, crowd sensing was performed using chat bot agents. Up to 200 digital twins (triangles) retrieved form crowd sensing surveys (chat dialogues, see Example 1) were added to the simulation dynamically. The *S* vector and the social distance *r* of digital twins now depend on the answers given by the (real) humans, which can differ from the initial *S* setting. These agents (if their *S* differs from the basic model) create a disturbance in the segregation patterns. Agents with *s*_αβ_ = 0/1 and *s*_αα_ = 0/1 can be integrated into both groups and are able to bind different groups close together (see the development and movement of the blue and red clusters inside the red circle in Figure 10). The figure shows different simulation snapshots at 500, 1000, and 1500 simulation steps. Each simulation step corresponds to a real-time interval of 10 s (to enable crowd sensing in real-time). The survey participants were chosen randomly by the chat bot agents.

Using an initial unified stable parameter set for all agents, digital twins with varying and different parameter sets based on survey data can be used to test the stability and convergence criteria of structure formation or mobility patterns like in traffic management on a fine-grain scale.

**Example** **1.**
*A typical survey job script for the Sakoda model.*


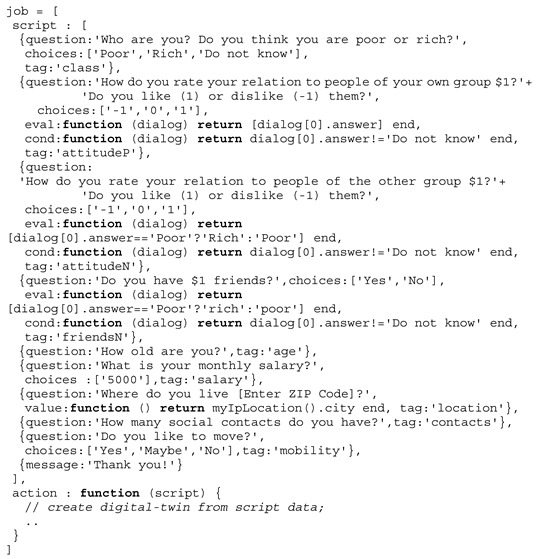



#### 9.1.2. Short-Term Mobility

The experiment aims to study crowd formation in cities (places, streets, and buildings) based on physical and virtual social interaction via social media and opinions and were performed with an artificial map of a city (simplified real world). Distributed crowd sensing via chat bots introduce updates and disturbance into the social formation structures. The mobility of physical agents is modelled by the extended Sakoda model introduced in Section 7 and depends on local social interaction (temporary crowd formation) and a potential field attraction regarding the destination in the city that has to be reached. Mobility is spatially constrained (streets).

The crowd sensing tries to estimate the local people’s movement and compound crowd formation constrained by urban structures (like streets) based on an estimated individual *S* matrix. For the sake of simplicity, two groups are assumed (*a*,*b*), but members of each group differ in attitudes and behaviour.

The two groups, for instance, could be car and bicycle users acting as public traffic participants, differing in social, mobility, and group formation behaviour. The survey can be opportunistic, e.g., ad-hoc and occasional with a situation-aware dialogue for specific traffic situations, or participatory with a more general survey character.

The position of the digital twins added to the two-dimensional simulation world is estimated by GPS and IP localisation collected by the chat bot agent or estimated by user answers.

Mobility constraints by streets were added, shown in Figure 11. The constraints are synthetic. The mobility of agents was driven by reaching a destination in the simulation world (either chosen randomly or by user information in the case of digital twins) and damped or delayed by social interaction, depending on individual *S* vector, social, and mobile interaction distances. The simulation was carried out with 400/400 class *a*/*b* agents.

The results (snapshots after 3500 simulation steps) show an increasing disturbance of social aggregation patterns resulting from social interaction (mobility due to social attraction) by goal driven traffic mobility (attraction by destination). Again, *s*_αβ_ = 0/1 and *s*_αα_ = 0/1 twins act as compound glue and bring homogenous groups closer together. With the increasing influence of goal-driven traffic mobility over social attraction and mobility, the social clustering gets fuzzier and more diffuse, shown on the right simulation snapshot of Figure 11.

#### 9.1.3. Evaluation

In both examples (with respect to long- and short-term mobility), digital twins with varying parameter sets have a significant impact on crowd structures and group behaviour. The digital twins pose individual behaviour not initially considered in the synthetic simulation. The behaviour is based on crowd sensing surveys performed by real humans. Short-term mobility and interaction based on the Sakoda model can be used in smart city crowd behaviour prediction.

### 9.2. Smart City Light Management

The goal of this use-case is to provide a simple demonstrator that uses simulation to investigate crowd interaction with smart city infrastructures by implementing a self-organised control of ambient light conditions (e.g., in streets or buildings). Crowd sensing provides real-world feedback as input for the augmented simulation to evaluate and predict real-world changes, finally used for decision-making processes (light management) [24]. The crowd gets feedback from the simulation by the environmental light control (output of the simulation).

#### 9.2.1. World

This simple demonstrator consists of a simulation containing an artificial city area with mobile physical agents representing humans interacting with mobile devices or machines), beacons (access points) for sensor aggregation and distribution, and some external beacons connected to the Internet enabling the connection to mobile devices via the Internet, and finally smart light devices illuminating streets and buildings, shown in Figure 12.

#### 9.2.2. Interaction and Control Model

The interaction model is rather simple and purely algorithmic in this use-case. The goal of the simulation is to control and predict light illumination in city areas based on activity, requirements (constraints), and people’s feedback retrieved by crowd sensing.

#### 9.2.3. Crowd Sensing

The goal of explorer agents (sent out by mobile or light devices) is information mining in the outside world via Internet deployed agent processing platforms that can be accessed by simulation nodes and WEB browser Apps. The collected information is passed back to the root node (e.g., a mobile device of a passenger) to assist decision making and navigation.

The explorer agents have to estimate the position of the root node by performing sensor mining from surrounding devices they are visiting and from the outside world (far away) by asking questions answered by humans via the WEB App chat dialogue. An example dialogue of the explorer agent is shown below in Example 2. The explorer asks a user for its current place and location within a region of interest and an assessment of the current light situation. Depending on the answer a specific action is suggested.

**Example** **2.**
*Survey job defining a dialogue for an explorer agent that migrates to mobile devices (smart phones) to participate in a chat dialogue with the user of the mobile device. The aim is to get environmental perception. Finally, an action function evaluating the survey is executed to deliver data.*


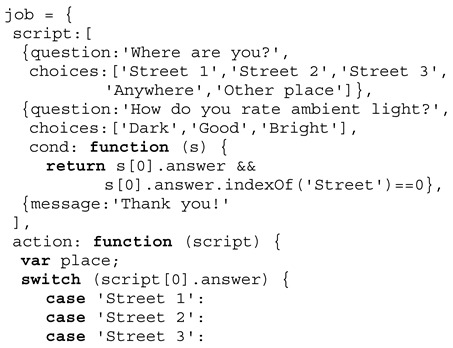



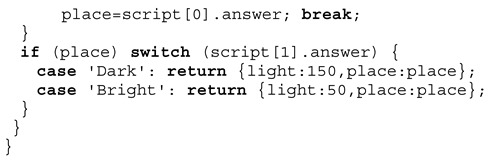



Based on answered questions regarding the current user location, the satisfaction of ambient light condition, and an optional fusion with device sensor data (light, position, etc.), actions are directed to smart light control devices to change the light condition in streets and buildings by using mobile notification agents. The action planning is based on crowd demands and energy-saving constraints. If action is required, mobile notification agents are sent out to neighbouring nodes to change light intensity based on directed diffusion, random walk, and divide-and-conquer approaches.

#### 9.2.4. Evaluation

The simulation with the *SEJAM* simulator was able to be performed with more than 1000 nodes, hundreds of beacons, and more than 10000 explorer agents. Real-time values for one simulation step depend on the number of active agents to be processed, node and agent mobility (graphics and communication), and ranges from 1 ms to 1 s. For this use-case, a real-time simulation step of 1 s is more than sufficient to control lights in streets and on places (or within buildings). The minimal response time from sensor input to actuator control is about 5 s.

The user feedback from crowd sensing surveys performed by mobile agents is used to optimise illumination in public areas like streets with respect to user and energy management constraints. Energy saving up to 30% can be achieved, light pollution can be decreased, and citizen satisfaction can be optimised. Combined with machine learning, the simulation (mapping a real-world city region) can be used to predict future light demands based on crowd sensing and using other physical sensors (e.g., crowd and traffic flow sensors).

### 9.3. Smart City Traffic Management

The goal of the traffic simulation is the recognition of flows, jams, and structures (clusters) on streets and places. In [21], adaptive and partly self-organising traffic management was achieved by using a Multi-agent System (MAS) with multi-levels of decision making and a hierarchical organisational structure.

The simulation results can be finally used to manage real-world traffic by agents in real-time. Real-world traffic flows can be influenced by technical traffic infrastructure, by traffic information systems, by vehicle driver behaviour (which can be diverse), or by influencing people’s decision making (on a temporally and spatially short-range scale)

#### 9.3.1. World

In general, the simulation world consists of a graph *G* = (*P*,*S*) with nodes *P* representing places or spatial regions and edges *S* representing streets. Here, the world consists of a simplified street map arranging streets orthogonally in a regular grid, although the traffic simulation is not limited to any specific map or graph geometry. An example street map is shown in Figure 13. Streets have a major orientation (horizontal or vertical) and consist of three tracks (ride, left, and middle). A vehicle can move in directions north (*N*), south (*S*), east (*E*), and west (*W*).

#### 9.3.2. Interaction and Mobility Model

The model parameters of vehicles and their driver agent or passengers (digital twins) are defined in Definition 3. They can be set with default values, by Monte Carlo simulation, or they can be selected and derived from crowd sensing surveys from real humans (in real-time).

**Definition** **3.**
*Typical model parameters of drivers and automatic vehicles.*


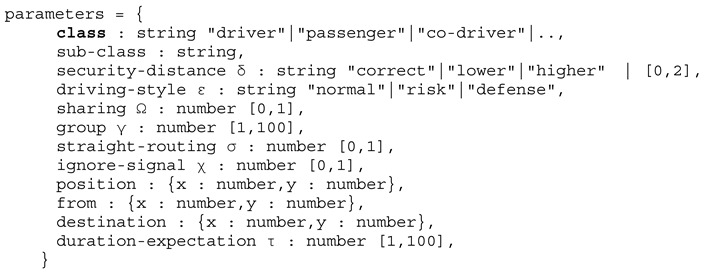



The *sub-class* parameter set can address specific classes like gender, age, or more general attributes like citizen status, or living conditions. The *security-distance* behaviour (between vehicles), the *driving-style* parameter, and the *straight-routing* parameter (for route planning) are only relevant for the *driver* class. A *straight-routing* = 0 parameter results in a zig-zag routing, otherwise a longer straight-line routing is used following the current street as long as possible (range depends on parameter value). The *ignore-signal* parameter is a probability to ignore stop signals (red signal), e.g., if the signal state changes.

A typical survey job script is shown in Example 3, showing a job being passed to a crowd sensing agent. It is used to derive the actual position, e.g., of a mobile device, motion, and to get information about the user (e.g., passenger or driver). Finally, a parameter set for the behaviour model is derived and digital twins are created.

**Example** **3.**
*A typical crowd sensing script job (JavaScript) for traffic flow exploration passed to a crowd sensing agent.*


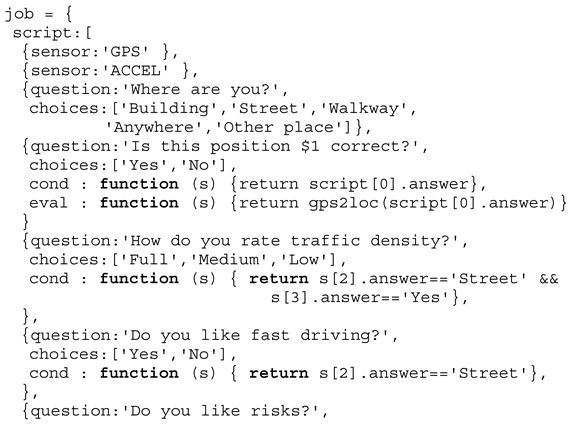



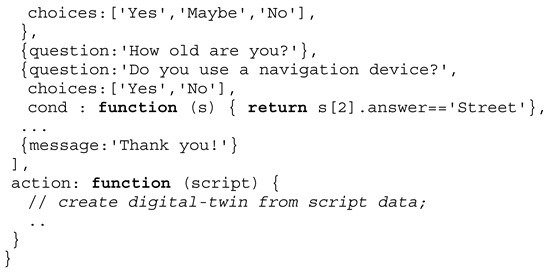



#### 9.3.3. Multi-agent System

The MAS in the simulation consists of a world agent that controls the simulation and establishes the connection to the outside world (Internet), vehicle, and twin agents. The twin agent can either be the active controller of the vehicle agent (influencer) or a passive passenger. The vehicle agents implement different overlaid behaviour:Constraint-based traffic mobility following specific rules (generalised behaviour)Social interaction (individual behaviour)Route-planning and optimisation (goal-based individual behaviour)

The vehicle agent consists of a set of activities performing specific actions and represent different micro-goals, basically: {*percept*, *plan*, *move*, *collect*, *wait*}. A vehicle agent represents a mobile entity and at least one human (a twin agent). The *collect* activity is able to bind more vehicle or twin agents (humans) to this vehicle entity, i.e., implementing vehicle grouping or sharing. Twin agents can be coupled to vehicle and communicate with each other via the tuple space of the mobile platform both sharing.

A vehicle has a direction *dir* and uses a set of sensors {*f*,*l*,*r*,*b*} providing information about the neighbourhood in front, left, right, and backward relative to current direction. Each sensor is an array with distance increasing with the array index. New sensor values are collected in the *percept* activity, shown in Algorithm 3.

**Algorithm 3.** Perception activity of the vehicle agent. Sensors k0, v0, and i can depend on twin driver agents, too.1k0 ← update keep going probability (twin)2**do** update sensors {p,s,d,v,i,f,l,r,b}:3p ← get current position (x,y)4  s ← get street at this position5  d ← compute distance to destination6  v ← compute speed7  v0 ← update highest speed8  i ← compute default security distance (s,v)9  **do in** current direction:10  f ← get front resources {street,vehicle}11  l ← get left resources {street,vehicle}12  r ← get right resources {street,vehicle}13  b ← get back resources {street,vehicle}

The vehicle mobility and its planning are immediately influenced by its velocity *v*, the sensors *d* and *i*(*s*,*v*) limiting the distance to the next vehicle, and the force to keep the current direction (not turning) *k*. There is a current long-range direction *dir* and a short-range position correction Δ, an (x,y) delta vector, e.g., required for turns or street and direction changes.

The current street is given by the *s* sensor. A street *s* consists of coordinates {*x*,*y*,*w*,*h*}, a unique identifier *id* and a set of possible directions *dirs*. It is assumed that a street has three tracks (left, right, middle). A street has either a horizontal (*N*,*S*) or vertical alignment (*W*,*E*). *P*(*x*,*y*) is place in the world at position (*x*,*y*) with attributes *street*, *vehicle*, *signal*.

In automatic driving mode, the distance to a vehicle in front is limited to *d* = *v*^δ^/*dk* and depends on the security distance parameter δ and the current velocity of the vehicle.

The vehicle plans the next movement based on perception in activity *plan*, shown in Algorithm 4 (simplified pseudo code), finally executing the movement in activity *move*, shown in Algorithm 5.

A twin agent is divided in two sub-class behaviour: (1) Driver (2) Passenger. The twin agents implement basically the goal-driven behaviour of mobility and provides higher planning and routing levels influencing the vehicle. A driver twin agent directly controls traffic routing, whereas passenger twin agents influence the temporal flow of vehicles (by entering or leaving the vehicle).

**Algorithm 4.** Simplified planning activity of the vehicle agent, which can be depend on twin driver and partly on passenger agents, too1**do** find **new** direction and position correction:2**if** Δ = 0 **then**3  //*0. Check correct direction*4   **if** k = 0 ∧ (dir∉s ∨ p∉{(x,y):(x,y)∈s ∧ on right side **of** dir}) **then**
5   (dir,Δ)←new dir(s,p)6   //*1. Check traffic signal (red?)*7   stop ← r[1].signal = red ∧ r[1].signal.dir = dir8   //*2. Navigate to destination*9   **if** k<k0 **then**
10    **case** dir **of**11    N => **if** p.x+ε < dst.x ∧ f[1].street≠s **then** next←E,Δ←(0,−1)12       **if** p.x−ε > dst.x ∧ f[1].street≠s **then** next←W,Δ←(−3,−3)13    S => **if** p.x+ε < dst.x ∧ f[1].street≠s **then** next←E,Δ←(3,3)14       **if** p.x−ε > dst.x ∧ f[1].street≠s **then** next←W,Δ←(−1,1)15    W => **if** p.y+ε < dst.y ∧ l[3].street≠s **then** next←S,Δ←(−2,3)16       **if** p.y−ε > dst.y ∧ r[1].street≠s **then** next←N,Δ←(0,−1)17    E => **if** p.y+ε < dst.y ∧ r[1].street≠s **then** next←S,Δ←(0,1)18       **if** p.y−ε > dst.y ∧ l[13].street≠s **then** next←N,Δ←(2,−3)19    **if** next **then** dir←next,k←k0+320   //*3. Check road crossing; keep dir.*21   **if** ¬next ∧ f[1].street≠s **then** k←k+422  **else if** blocked ∧ random(1) **then**23   //*4. Escape blocking situation*24   **do** ∀ dir ∈ {N,S,W,E}:25    find one neighbour place q **of** p26     **with** |q−p] = 1 ∧ street∈q = s ∧ vehicle∉q ∧ Δ.x/y = 0:27        Δ←Δ+(q−p)

**Algorithm 5.** Simplified move activity of the vehicle agent. The gotoXY function moves the vehicle one step to a new free place minimizing minΔ→01go ← true2//*1. Check free distance*3
4  **do** ∀ n **in** {1,2,,..i}:5   **if** f[n].vehicle **then** go ← false6  **if** Δ = 0 ∧ go **then**
7   **case** dir **of**
8    N => next←{x:p.x,y:p.y−1}9    S => next←{x:p.x,y:p.y+1}10    W => next←{x:p.x−1,y:p.y}11    E => next←{x:p.x+1,y:p.y}12   **if** vehicle∉P(next) **then** blocked←0,goto(next)13   **else if** v = 0 **then** incr(blocked)14  **else** Δ←gotoXY(Δ,p)

#### 9.3.4. Evaluation

The traffic simulation was initially performed with a closed simulation world using a default behaviour model and traffic control parameter set. The outcome of this simulation was compared with an open simulation using agents with the same default behaviour extended with digital twins created from user surveys via crowd sensing agents. The major goal was to show the influence of variation by real-word digital twins on the simulation outcome and to identify relevant model parameters with significant impact. The traffic flow was controlled by light signals (red and green signals) positioned at each crossing corner. The default traffic signal switching is periodic with a state sequence {rr,gr,rr,rg}, with the first character assigned to all North-South signals, and the second character assigned to all West-East signals (r:red/g:green). Transition times were *t*_1_ for gr|rg → rr transitions, and *t*_0_ < *t*_1_ for rr → gr|rg transitions.

In Figure 14, a closed simulation with vehicles (individual class) and different traffic signal switching times were investigated. The simulation started with an initial population of 200 vehicles at random starting positions. The plots show the time-resolved decrease of driving passengers and increase of arriving passengers (in this case vehicles with on driver as a passenger) due to arrival at a destination position. The first plot (a) uses a signal switch time (red ↔ green signal state) of 500, the second plot of 200 simulation time units. Either North-South (*NS*) streets get green lights, or the West-East (*WE*) streets. The switching time has a significant impact on the mean arrival time of vehicles, although vehicles uses *NS* and *WE* streets, but the arrival time do not depend on individual parameters, i.e., security distance and routing strategy (zig-zag versa straight-line). The vehicle flow depends on the routing behaviour (straight-line shows lower dependence on signal switching times).

In Figure 15, the simulation was extended with MCWS and digital twins posing individual behaviour. The MCWS users were chosen randomly. The goal of this simulation was to identify jams and the dependence on driver behaviour parameter. Two main parameters were identified: The security distance and routing (straight-line vs. zig-zag shortest path) behaviour. Again, about 200 vehicles were created, a fraction with a default behaviour, and another fraction with individual behaviour. The value of the security distance parameter δ has significant impact on the jam probability (defined by the number vehicles involved in jams relative to the overall number of vehicles). A value δ = 1.0 relates to a law-conforming driving style, a lower value means a riskier driving style with lower distances to other vehicles, and a higher number a more defence driving style. Lower security distances (relative to current vehicle velocity) increases the jam probability due to a modified vehicle flow (accidents are not considered here). The is more important if a straight-line routing style (parameter greater than zero) is chosen by the driver. The last three bars in the plot show a distribution of the security-distance parameter (25%, 33%, 50%). Even a low fraction of 25% risky drivers increases the jam probability significantly.

Finally, the traffic control algorithm was evaluated in Figure 16. Again, the traffic flow of 200 vehicles with zig-zag routing was evaluated (δ = 1.0, σ = 0). Two traffic signal control algorithms were compared: On the left side the static global switching algorithm (with *t*_1_/*t*_0_ = 500/60), and on the right side a dynamic local switching algorithm with dynamic switching times based on vehicle flow densities before each signal of a crossing. The global mode switches all signals of crossings globally and periodically, the local mode switches signals of each crossing individually. The dynamic flow-based switching increases arrival times by 2 times, and the velocity of vehicles shows low modulations and is near by the default velocity *v* = 50 (arb. units).

To summarise, individualism of behaviour model parameters derived by real-world sensing (eventually in real-time) like driving style (security distance, routing) can influence crowd and vehicle flows significantly and hence has to be considered by traffic control to optimise traffic and crowd flows. It is important to get a representative model parameter set from an actual traffic situation to be able to predict flows and to control traffic.

## 10. Conclusions

The MAS simulation framework presented in this work provides the bidirectional integration of real-world data via agent-based crowd sensing, agent-based modelling, and agent-based simulation via data mining, i.e., applying DM in ABMS by creating generated simulation data and applying ABMS in DM by backpropagating generated simulation data. The framework is suitable to combine social and computational simulations with real-world interaction at run-time and in real-time by using mobile computational agents. The simulator supports two classes of agents. Physical agents only exist in the virtual world and correspond to *NetLogo* agents, whereas computational agents can exist in real and virtual worlds and create the bridge between the two worlds.

The *JAM* agent processing platform is the core component in the simulator and crowd sensing system that can be deployed in heterogeneous networks on a broad range of devices including simulation software. This is enabled by programming *JAM* and agents in *JavaScript*. Chat bots with dynamic (situation and context-dependent) dialogues can be implemented with *JAM* agents directly using a unified dialogue structure and API.

Two key concepts in the augmented simulation approach are parametrizable behaviour models and digital twins created from surveys via crowd sensing.

Different use-cases demonstrate the augmented simulation approach with social or socio-technical interaction models. First, a classical social interaction study was performed based on a parametrizable Sakoda model, second, a reactive and interactive city light management was sketched, and third, a crowd-based city traffic control was investigated. Digital twins retrieved by agent-based crowd sensing extended agent-based simulation with variance and a broader range of model parameter settings. The crowd sensing performed by mobile agents is used to create digital twins of real humans (with respect to the social interaction model and mobility) based on individual surveys via a chat bot dialogue. Chat bots are the link between virtual and physical worlds.

It could be shown that social interaction and individualism (variance) has a relevant impact on traffic control and crowd or vehicle flows, hence social models overlay technical models. It is important to get a representative model parameter set and distributions from real-world situations by crowd sensing for accurate system prediction and control.

In the future, we aim to use the framework in order to investigate different environments in quasi-experimental set-ups with broader data and user range.

Finally, the system behaviour and cross interaction effects should be studied with hybrid models, i.e., the combination of the social interaction model describing group formations, e.g., Sakoda model, with mobility models, e.g., mobile vehicles, or by combining agent-based with analytical models to strength the simulation output.

## Figures and Tables

**Figure 1 sensors-19-04356-f001:**
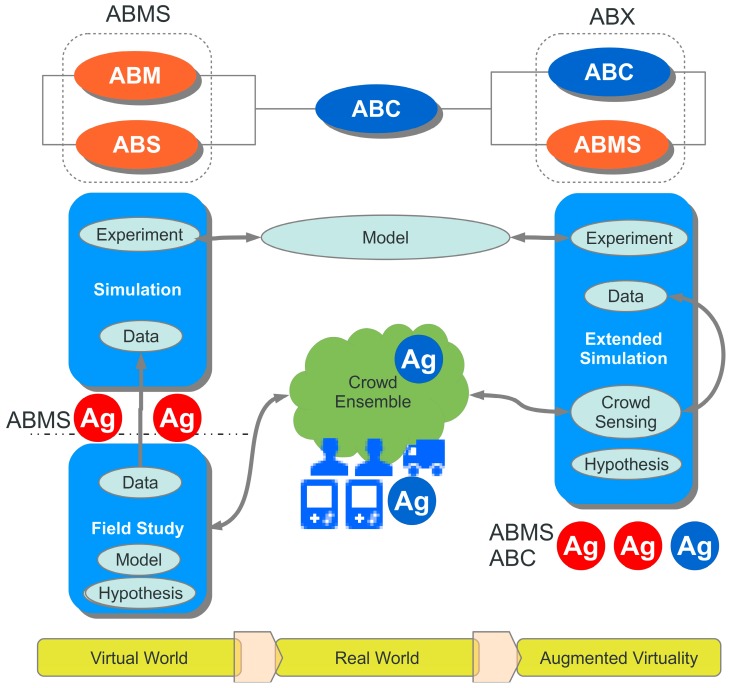
(**Left**) Traditional field studies and agent-based simulation of social systems. (**Right**) New combined agent-based simulation and agent-based crowd sensing enabling bidirectional data exchange.

**Figure 2 sensors-19-04356-f002:**
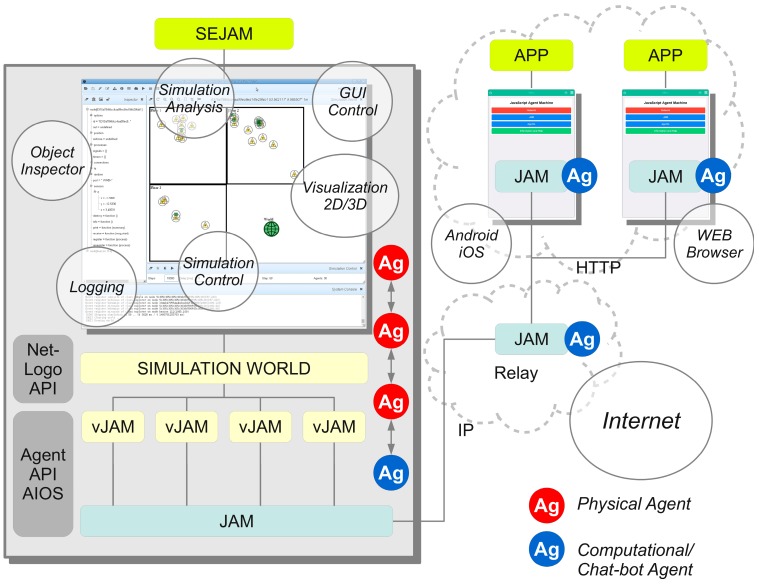
The principle concept of closed-loop simulation for augmented virtuality: (**Left**) Simulation framework based on the JavaScript agent machine (JAM) platform (**Right**) Mobile and non-mobile devices executing the JAM platform connected with the virtual simulation world (via the Internet) [24].

**Figure 3 sensors-19-04356-f003:**
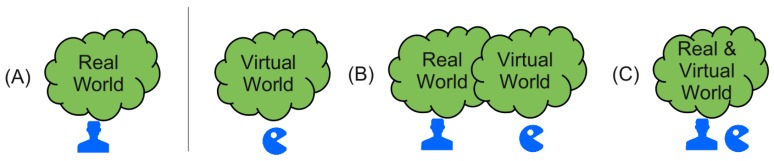
(**A**) Real-world only deployed with humans. (**B**) non-overlapping real and virtual world, and (**C**) overlapping real and virtual world.

**Figure 4 sensors-19-04356-f004:**
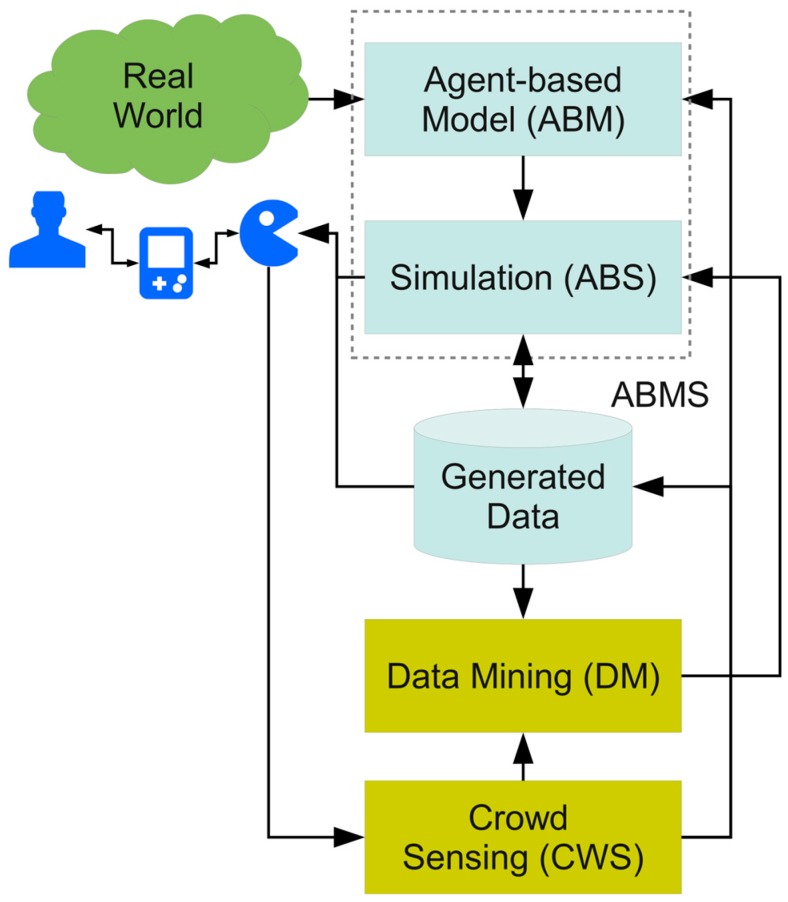
Principle work and data flow integrating agent-based crowd sensing in agent-based modelling and simulation.

**Figure 5 sensors-19-04356-f005:**
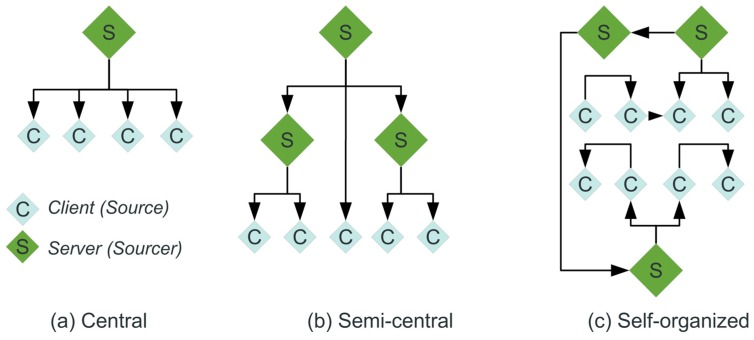
Different crowd sensing architectures and strategies with data sourcer and data sources.

**Figure 6 sensors-19-04356-f006:**
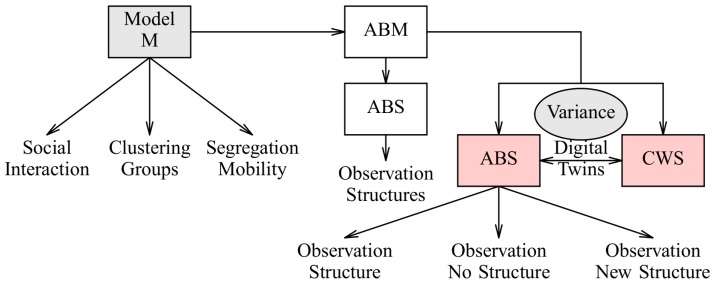
Crowd sensing combined with agent-based simulation (ABS) adds variance to the synthetic simulation world by digital twins.

**Figure 7 sensors-19-04356-f007:**
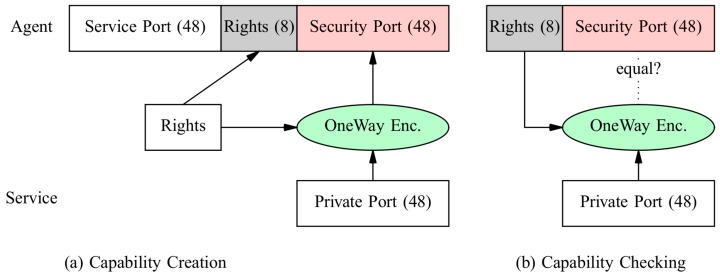
(**Top**) Capability format. (**a**) Capability creation by encoding private security port and new rights field. (**b**) Capability checking by encoding requested rights field and private security port and comparison with provided security port.

**Figure 8 sensors-19-04356-f008:**
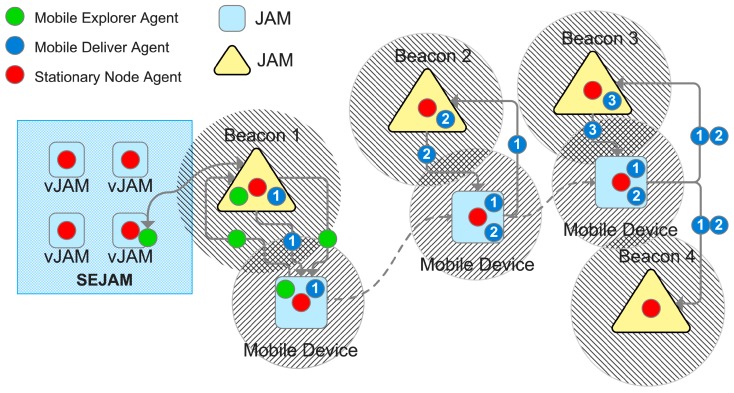
Virtual simulator world connected to spatially distributed non-mobile beacons, mobile devices connected temporarily to beacons (cellular or local WIFI networks), and mobile agents (crowd sensing and chat bots) used for wide-range interaction.

**Figure 9 sensors-19-04356-f009:**
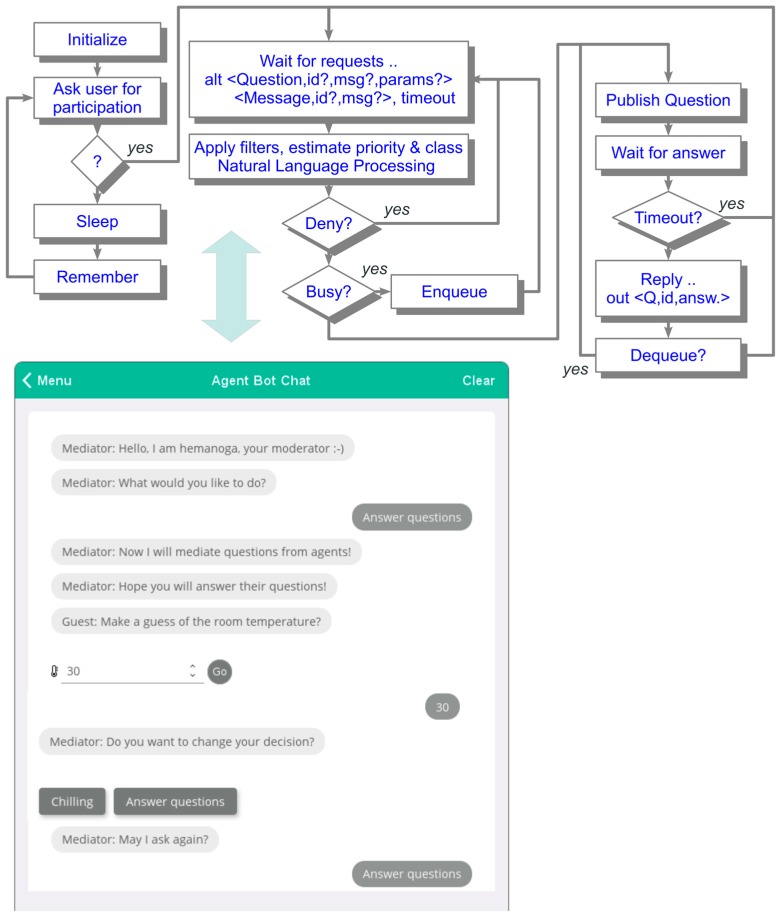
(**Bottom**) Chat dialogue of the JAM App showing messages and questions from chat bot agents and principle mediator agent behaviour scheduling the user chat (**Top**).

**Figure 10 sensors-19-04356-f010:**
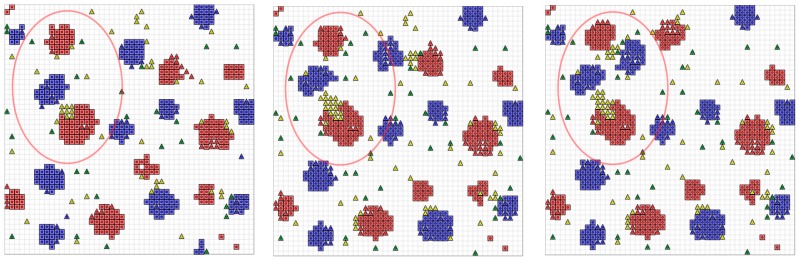
Simulation world with long-term mobility, without spatial and context constraints at different simulation times (500/1000/1500 steps) consisting of 200/200 a/b class agents (blue/red squares) all with S = (1,−1,−1,1) and r = 3 parameter settings and additionally up to 200 digital twins (triangles with colour based on individual S/r parameters).

**Figure 11 sensors-19-04356-f011:**
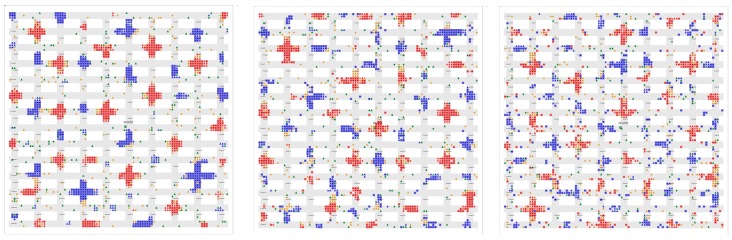
World consisting of mobility constraints (streets), 400/400 class a/b agents (blue and red squares) and digital twins (triangles) with different model parameters. (**Left**) Only social-driven mobility and clustering. (**Middle**) Social and traffic driven mobility with low attraction of the destination. (**Right**) Social and traffic driven mobility with high attraction of the destination.

**Figure 12 sensors-19-04356-f012:**
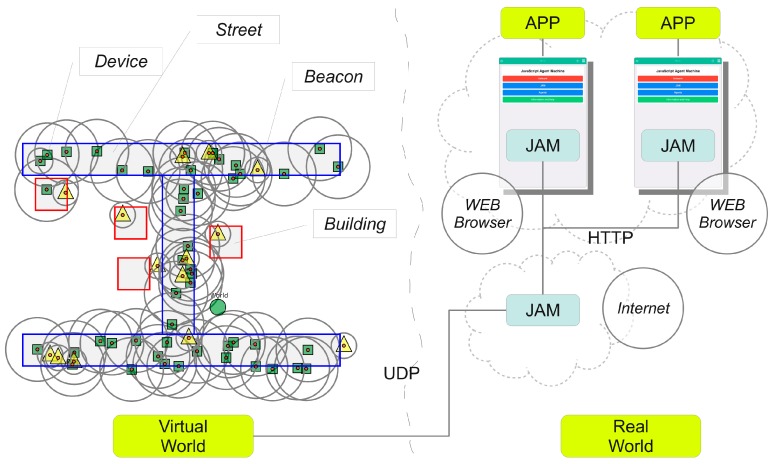
A demonstrator: An artificial street area consisting of street segments (blue rectangle), buildings (red rectangles), and beacons (yellow triangles), populated with stationary and mobile devices (green squares). The grey circles around beacons and nodes show the wireless communication range. Only overlapping circles connect.

**Figure 13 sensors-19-04356-f013:**
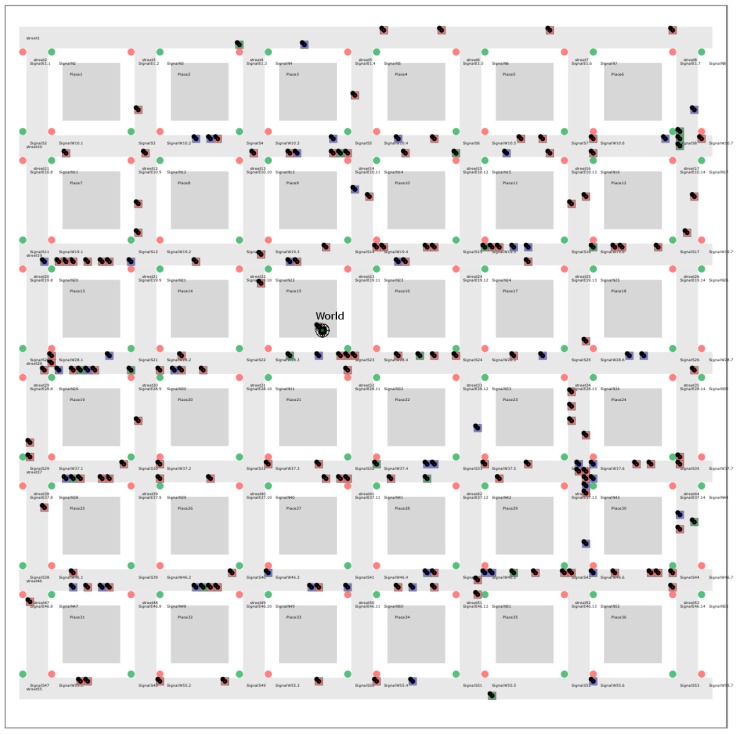
A typical simulation world consisting of a street map, traffic signals, places, and different vehicle classes (red: individual, blue: shared, and green: public).

**Figure 14 sensors-19-04356-f014:**
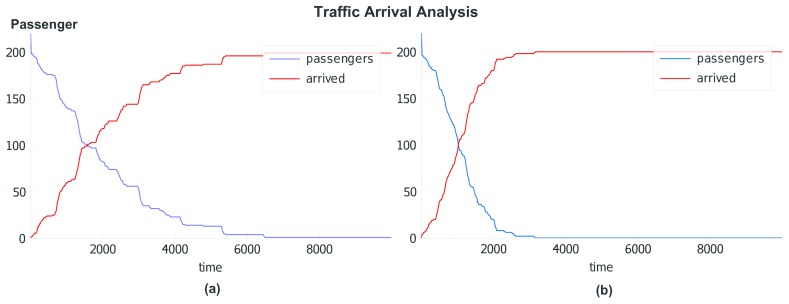
Simulation with 200 vehicles with static traffic control showing the accumulated number of moving vehicles/passengers and the number of arrived vehicles/passengers (**Left**) t_1_/t_0_ = 500/60 signal change times (**Right**) t_1_/t_0_ = 200/60.

**Figure 15 sensors-19-04356-f015:**
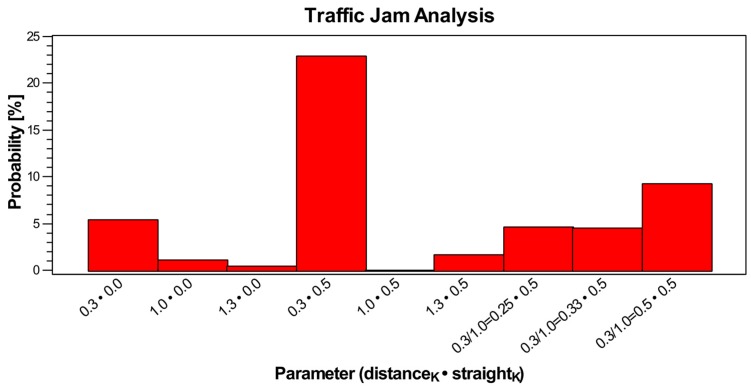
Simulation with 200 vehicles and different mobility parameters showing the probability of a jam.

**Figure 16 sensors-19-04356-f016:**
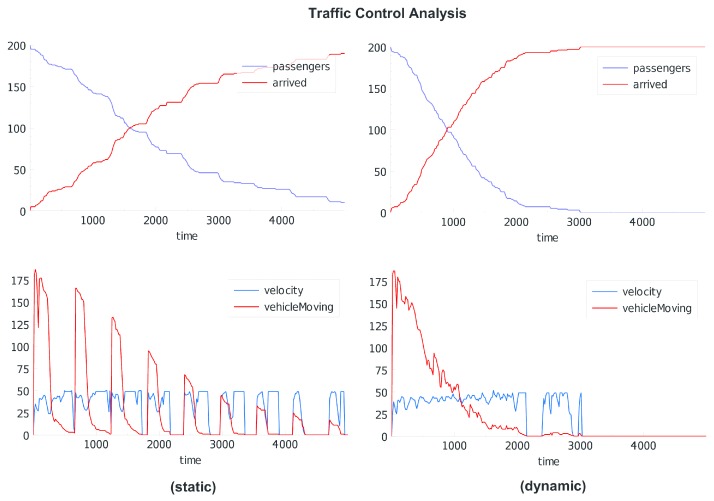
Simulation of 200 vehicles (σ = 0.0,δ = 1.0): (**Top**) Temporal development of number of mobile vehicles/passengers and number of vehicles/passengers reached destination (**Bottom**) Mean velocity of moving vehicles and number of moving vehicles (**Left**) Without traffic control with t_1_/t_0_ = 500/60 signal change times (**Right**) With dynamic flow traffic management.

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
