# Peer review of "Real-Time Human-In-The-Loop Simulation with Mobile Agents, Chat Bots, and Crowd Sensing for Smart Cities"

_sensors, 2019, doi:10.3390/s19204356_

Round 1
Reviewer 1 Report
Please find comments in the attached txt file

Reviewer 2 Report
The authors designed a real-time social network simulation framework. Such simulation framework could interact with real world environments and conduct crowdsensing work via a chat blog. The authors conduct three experiments to validate the practicability of their framework.
Although lots of works have been done, some drawbacks still exist.
The motivation is not clear. The reviewer is not clear not why the simulation system is important and significant for the real world practice. The authors seem only combine some simple system into one. Moreover, the contribution is also not clear. It is suggested that the authors could make them more clear. The organization is poor. The organization of this paper is messy and makes it hard to read. Some of the figures are also not clear. Moreover, numerous grammar mistakes exist throughout the paper. The authors need carefully check their paper and correct all of them. The simulation part is not convincing. 1) The detailed settings are not provided. The authors only state some simple experimental settings under the figures. 2) The comparisons with other simulation framework are not conducted, instead of some statements by words, which are not enough. The authors failed to properly cite several past literatures (e.g., [1-4]) highly related to this work.[1] Dejun Yang, Guoliang Xue, Xi Fang, and Jian Tang, “Crowdsourcing to Smartphones: Incentive Mechanism Design for Mobile Phone Sensing”,in ACM International Conference on Mobile Computing and Networking (MobiCom), 2012.
[2] Haiming Jin, Lu Su, Danyang Chen, Klara Nahrstedt, Jinhui Xu, "Quality of Information Aware Incentive Mechanisms for Mobile Crowd Sensing Systems", the 16th ACM Symposium on Mobile Ad Hoc Networking and Computing (MobiHoc 2015), Hangzhou, China, June 2015.
[3] Liang Wang, Zhiwen Yu, Daqing Zhang, Bin Guo, Chi Harold Liu: Heterogeneous Multi-Task Assignment in Mobile Crowdsensing Using Spatiotemporal Correlation. IEEE Trans. Mob. Comput. 18(1): 84-97 (2019)
[4] Haiming Jin, Lu Su, Houping Xiao, Klara Nahrstedt, "Incentive Mechanism for Privacy-Aware Data Aggregation in Mobile Crowd Sensing Systems", IEEE/ACM Transactions on Networking (TON), Vol. 26, No. 5, Pages 2019-2032, August 2018.
Round 2
Reviewer 1 Report
In the revised version of the paper, the Authors extended comments and explanations about the proposed algorithm. However, an analytical model justifying the proposed solution is still missing. Thus, I think the paper should be revised again.
Author Response
See attached PDF

Reviewer 2 Report
The authors have addressed fairly well my comments to the previous version. I do not have any further comments and thus recommend that this paper be accepted.
Author Response
See attached PDF
